# A Notch positive feedback in the intestinal stem cell niche is essential for stem cell self-renewal

Kai-Yuan Chen[1,2,†] iD, Tara Srinivasan[3,†], Kuei-Ling Tung[4,†], Julio M Belmonte[5] iD, Lihua Wang[2,4], Preetish Kadur Lakshminarasimha Murthy[6], Jiahn Choi[2], Nikolai Rakhilin[1,2], Sarah King[3], Anastasia Kristine Varanko[4], Mavee Witherspoon[6], Nozomi Nishimura[3] iD, James A Glazier[5], Steven M Lipkin[7], Pengcheng Bu[2,8,*] iD & Xiling Shen[1,2,3,**] iD

## Abstract

The intestinal epithelium is the fastest regenerative tissue in the body, fueled by fast-cycling stem cells. The number and identity of these dividing and migrating stem cells are maintained by a mosaic pattern at the base of the crypt. How the underlying regulatory scheme manages this dynamic stem cell niche is not entirely clear. We stimulated intestinal organoids with Notch ligands and inhibitors and discovered that intestinal stem cells employ a positive feedback mechanism via direct Notch binding to the second intron of the Notch1 gene. Inactivation of the positive feedback by CRISPR/Cas9 mutation of the binding sequence alters the mosaic stem cell niche pattern and hinders regeneration in organoids. Dynamical system analysis and agent-based multiscale stochastic modeling suggest that the positive feedback enhances the robustness of Notch-mediated niche patterning. This study highlights the importance of feedback mechanisms in spatiotemporal control of the stem cell niche.

**Keywords** gene editing; intestine stem cell; Notch signaling; organoid; positive feedback

**Subject Categories** Quantitative Biology & Dynamical Systems; Signal Transduction; Stem Cells

**Mol Syst Biol. (2017) 13: 927**

## Introduction

The stem cell niche provides a spatial environment that regulates stem cell self-renewal and differentiation (Lander *et al*, 2012). One example is at the base of the intestinal crypt, where self-renewing LGR5$^+$ crypt base columnar (CBC) cells and lysozyme-secreting Paneth cells form a mosaic pattern in which each Paneth cell is separated from others by LGR5$^+$ cells (Barker *et al*, 2007; Sato *et al*, 2011a). As proliferative intestinal stem cells (ISCs), CBCs mostly divide symmetrically, compete with each other in a neutral drift process, and regenerate the intestinal epithelium in 3–5 days (Lopez-Garcia *et al*, 2010; Snippert *et al*, 2010). The stem cell niche is capable of recovering from radiation or chemical damages to restore tissue homeostasis (Buczacki *et al*, 2013; Metcalfe *et al*, 2014).

Regulation of the niche is a concerted effort involving various signaling pathways. Paneth cells provide niche factors including epidermal growth factor (EGF), Wnt ligands (WNT3A), Notch ligands, and bone morphogenetic protein (BMP) inhibitor Noggin to support CBC self-renewal, while pericryptal stromal cells underneath the niche also supply additional Wnt ligands (WNT2B) (Barker, 2014). Among the pathways, juxtacrine Notch signaling pathway is often linked to developmental patterning (Artavanis-Tsakonas *et al*, 1999; Kopan & Ilagan, 2009). Notch signaling is mediated through direct cell-to-cell contact of membrane-bound Notch ligands on one cell and transmembrane Notch receptors on adjacent cells. The extracellular domain of Notch receptors binds Notch ligands, which activates receptor cleavage that releases the Notch receptor intracellular domain (NICD) to translocate to the nucleus. NICD interacts with the DNA-binding protein RBPJk to activate expression of downstream genes, such as the HES family

1 School of Electrical and Computer Engineering, Cornell University, Ithaca, NY, USA
2 Department of Biomedical Engineering, Duke University, Durham, NC, USA
3 Department of Biomedical Engineering, Cornell University, Ithaca, NY, USA
4 Department of Biological and Environmental Engineering, Cornell University, Ithaca, NY, USA
5 Biocomplexity Institute and Department of Physics, Indiana University, Bloomington, IN, USA
6 School of Mechanical Aerospace Engineering, Cornell University, Ithaca, NY, USA
7 Departments of Medicine, Genetic Medicine and Surgery, Weill Cornell Medical College, New York, NY, USA
8 Key Laboratory of RNA Biology, Key Laboratory of Protein and Peptide Pharmaceutical, Institute of Biophysics, Chinese Academy of Sciences, Beijing, China
  *Corresponding author. Tel: +1 919 681 9184; E-mail: bupc@ibp.ac.cn
  **Corresponding author. Tel: +1 919 681 9184; E-mail: xs37@duke.edu
  †These authors contributed equally to this work

transcription factors. Notch signaling is essential for intestinal stem cell self-renewal and crypt homeostasis (Fre *et al*, 2005; van der Flier & Clevers, 2009). Among Notch receptors, inhibition of both Notch1 and Notch2 completely depletes proliferative stem/progenitor cells in the intestinal epithelium (Riccio *et al*, 2008). Inhibition of Notch1 alone is sufficient to cause a defective intestinal phenotype, while inhibition of Notch2 alone causes no significant phenotype (Wu *et al*, 2010). Notch3 and Notch4 are not expressed in the intestinal epithelium (Fre *et al*, 2011). Among Notch ligands, DLL1 and DLL4 are essential and function redundantly, and inactivation of both causes loss of stem and progenitor cells; in contrast, JAG1 is not essential (Pellegrinet *et al*, 2011).

In this study, we characterized the response of Notch signaling components in LGR5$^+$ CBCs from intestinal organoids and identified a direct Notch positive feedback loop. Perturbation to the positive feedback by CRISPR/CAS9 mutation of the binding sequence significantly reduced the number of CBCs in the stem cell niche. Computational modeling suggests that the positive feedback may contribute to robustness of the system when proliferation rates are high.

## Results

### Notch lateral inhibition and positive feedback

We characterized Notch signaling response in CBCs and Paneth cells (Fig 1A) using the *in vitro* intestinal organoid system (Sato *et al*, 2009), from which LGR5-EGFP$^+$ CBCs and CD24$^+$ Paneth cells were isolated using an established protocol (Sato *et al*, 2011a). Immunofluorescence (IF) confirmed that the sorted CD24$^+$ Paneth cells express lysozyme (Fig EV1A). RT–qPCR on purified CBCs and Paneth cells confirmed that Notch receptors (Notch1, Notch2) and signaling effectors (Hes1, Hes5) are enriched in CBCs, while Notch ligands (Dll1, Dll4, Jag1) and the secretory lineage regulator, Atoh1 (Yang *et al*, 2001), are enriched in Paneth cells, largely consistent with previous microarray measurements (Sato *et al*, 2011a; Fig 1B). Inhibition of Notch receptor cleavage by the γ-secretase inhibitor DAPT reduced the number of CBCs and increased the number of Paneth cells, whereas Notch activation by recombinant ligand JAG1 embedded in Matrigel (Sato *et al*, 2009; Van Landeghem *et al*, 2012; VanDussen *et al*, 2012; Yamamura *et al*, 2014; Mahapatro *et al*, 2016; Srinivasan *et al*, 2016) or EDTA (Rand *et al*, 2000) increased the number of CBCs and decreased the number of Paneth cells

(Fig 1C). Inhibition of Notch by DAPT up-regulated ligand expression, indicating that active Notch signaling suppresses ligand expression (Fig 1D and E). This is consistent with a lateral inhibition (LI) mechanism previously reported in several developmental systems, where ligands on a "sender" cell (in this case, Paneth cell) activate receptors on a "receiver" cell (in this case, CBC), which, in turn, suppresses ligand expression in the receiver cell (Collier *et al*, 1996). This intercellular feedback scheme causes bifurcation between adjacent cells, resulting in two opposite Notch signaling states (Fig 1F).

Additionally, Notch activation by recombinant JAG1 embedded in Matrigel (Sato *et al*, 2009; Van Landeghem *et al*, 2012; VanDussen *et al*, 2012; Yamamura *et al*, 2014; Mahapatro *et al*, 2016; Srinivasan *et al*, 2016) or EDTA (Rand *et al*, 2000) significantly increased receptor (Notch1/2) expression, while DAPT significantly reduced receptor expression (Fig 1D and E). This suggests the existence of a positive feedback loop, where activated Notch receptors up-regulate their own expression (Fig 1F).

### NICD directly activates Notch1 transcription

Although both Notch1 and Notch2 form positive autoregulation, Notch1 has a stronger response than Notch2 (Fig 1D and E). This is consistent with previous reports showing that Notch1 and Notch2 are somewhat functionally redundant, but Notch1 is more critical to stem cell self-renewal and crypt homeostasis, while Notch2 is dispensable (Wu et al, 2010). We performed lineage tracing using tamoxifen-inducible Notch1$^{CreER}$ × ROSA26$^{tdTomato}$ transgenic mouse reporter strains (Fre *et al*, 2011; Oh *et al*, 2013). After induction, labeled Notch1$^+$ cells showed a similar pattern that largely overlaps with CBCs in the niche (Fig 2A). From day 1 to day 3, marked progeny of Notch1$^+$ cells expanded out of the niche and overtook the trans-amplifying (TA) progenitor compartments; by day 30, the marked clones of the original Notch1$^+$ cells replaced the entire epithelium (Figs 2B and EV1B). These lineage tracing experiments confirmed that Notch1 is active in CBCs, which is consistent with previous findings (Wu et al, 2010; Fre *et al*, 2011; Pellegrinet *et al*, 2011; Oh *et al*, 2013).

We analyzed the LICR ChIP-Seq dataset of mouse small intestinal cells from ENCODE using the UCSC genome browser (Consortium, 2012) to investigate regulation of Notch1 and Notch2 transcription. The second intron region of the Notch1 gene is highly enriched with enhancer histone marks H3K4me1 and H3K27ac, while no such

**Figure 1. Notch levels in niche cells.**

A   Left: Cross-sectional view of murine intestinal crypt bottoms with co-immunofluorescence (co-IF) showing intermingled LGR5-EGFP$^+$ (green) CBCs and lysozyme$^+$ (LYZ, red) Paneth cells. Scale bar: 50 μm. Right: Schematic illustration of a niche pattern in both longitudinal and cross-sectional views of a crypt.

B   RT–qPCR quantification of Notch signaling components in CBC and Paneth cell populations. The experiment was performed in triplicate and presented mean ± SEM (***$P \leq 0.001$, **$P \leq 0.01$, *$P \leq 0.05$; Student's *t*-test).

C   Representative FACS plots of organoids treated with DMSO, JAG1 (embedded in Matrigel), EDTA or DAPT for 48 h, including gated analysis to isolate CD24$^{high}$/SSC$^{high}$ Paneth cells and LGR5-EGFP$^+$ CBCs according to an established protocol (Sato *et al*, 2011c).

D   RT–qPCR quantification of Notch signaling components in CBCs and Paneth cell populations after organoids were treated with Matrigel-embedded JAG1 (top), EDTA (middle), or DAPT (bottom). The experiments were performed in triplicate and presented mean ± SEM (**$P \leq 0.01$, *$P \leq 0.05$; Student's *t*-test).

E   Western blot analysis of Notch signaling components from conditions described in (D). Actin was used as a loading control.

F   Schematic illustration of lateral inhibition and positive feedback between neighboring cells. Transparent colors and dotted lines represent low expression/activity levels.

Source data are available online for this figure.

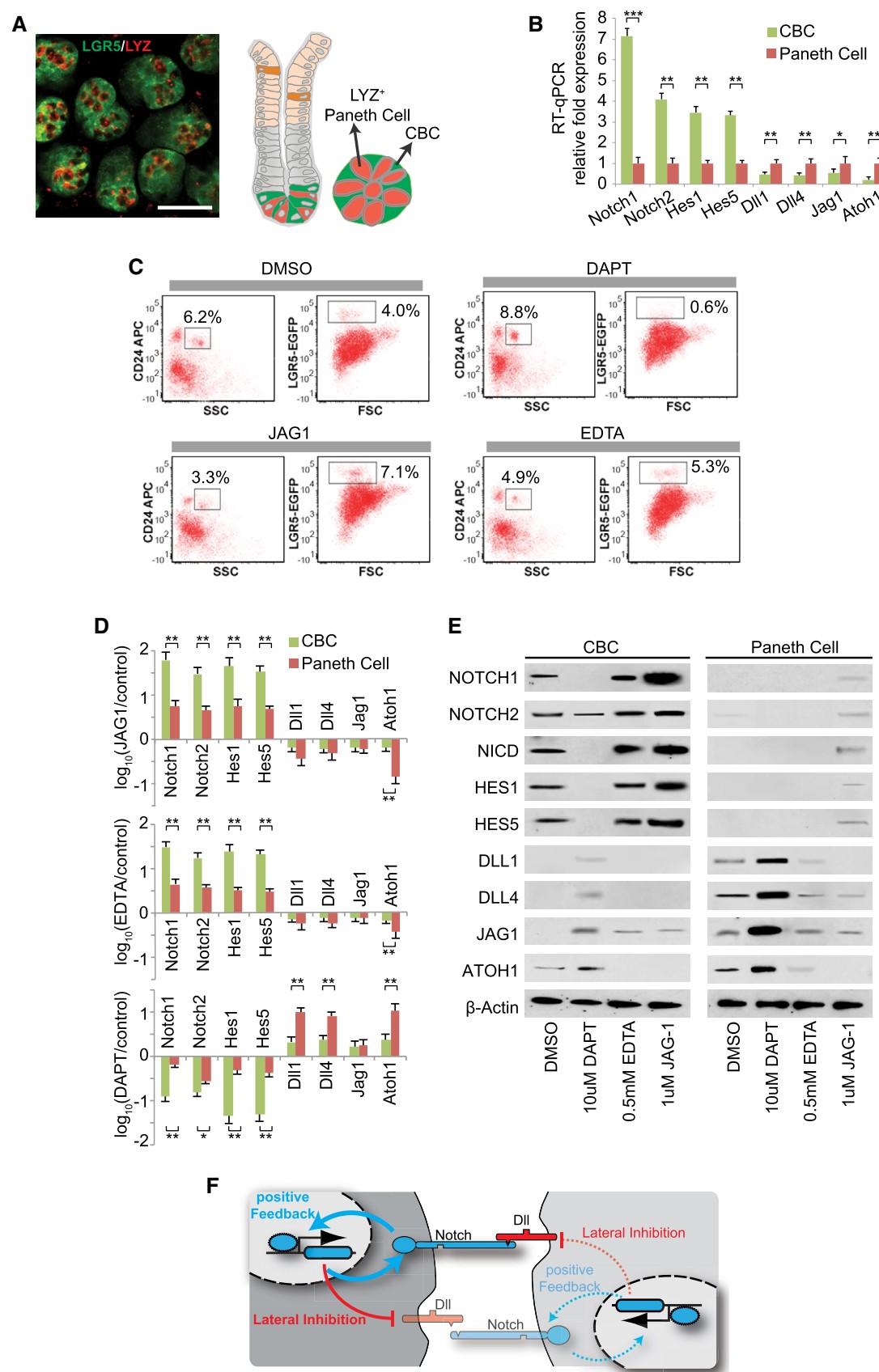

**Figure 1.**

regions were found in the Notch2 sequence (Fig EV2A). Computational analysis of this region with MotifMap (Wang *et al*, 2014) predicted a putative binding motif for RBPJk, the DNA-binding protein that forms an effector complex with NICD to activate Notch signaling. A unique eight base pair sequence (TTCCCACG, Chr2: 26,349,981–26,349,988) was identified (Fig EV2B). ChIP-PCR shows that NICD binds to this sequence in CBCs, and the binding was enhanced by JAG1 activation of receptors and suppressed by DAPT inhibition of receptor cleavage (Fig EV2C). For further validation, we crossed a LGR5-EGFP strain with a Rosa26-YFP-NICD strain (Oh *et al*, 2013) to generate a tamoxifen-inducible LGR5-EGFP-CreERT2 × Rosa26-YFP-NICD (NICD-OE) mouse strain. ChIP-PCR analysis of tamoxifen-induced NICD-expressing intestinal cells from NICD-OE mice also showed elevated NICD binding compared to uninduced control (Fig EV2D).

To further validate this enhancer sequence motif, we performed a luciferase reporter assay with the enhancer sequence cloned to the pGL4.27 [luc2P/minP] luciferase reporter vector containing a minimal promoter (Fig 2C). We compared the wild-type binding sequence with three mutated sequences (Fig 2C): (i) partial mutation of 3 nts in the binding sequence and 3nts of adjacent flanking region, (ii) partial deletion of 2nts of binding sequence, and (iii) mutation of the entire 8 nt binding sequence. The luciferase reporter vectors were transfected into intestine cells directly isolated from the mouse intestine with a pRL-SV40 control vector containing no binding sequence. The luciferase signal from the wild-type enhancer sequence was significantly higher than those from the mutated sequences or the control vector (Fig 2C). Jag1 and DAPT treatments also elicited stronger responses from the wild-type sequence than the mutated sequences or the control vector (Fig 2C).

We then performed a pull-down assay to confirm interaction between NICD and the identified binding motif. Oligonucleotides of the wild-type and mutated enhancer sequences were synthesized and labeled with biotin to pull down NICD/RBPJk complex from mouse intestinal crypt lysates, which was validated by Western blot (Fig 2D). The wild-type sequence has stronger NICD binding than the mutated sequences.

## Positive feedback is critical to self-renewal, niche homeostasis, and recovery

Our characterization of Notch signaling pathways in niche cells suggests that both LI and a direct positive feedback are active. The role of LI in the niche can be easily rationalized, because LI is known to regulate developmental patterns (Collier *et al*, 1996; Kim *et al*, 2014). However, it is unclear whether the Notch positive feedback has any function. CRISPR-Cas9 vectors were designed to target the NICD binding sequence (Fig 2E, Table EV1). CRISPR/Cas9 vectors with specific guide RNAs (gRNAs) were transfected into single LGR5-EGFP CBCs, which were subsequently propagated as organoids. After puromycin selection, individual colonies were picked and sequenced separately to confirm CRISPR/Cas9 editing in the cells. Sequencing results indicate the presence of indels in the target NICD binding region formed through non-homologous end joining (NHEJ) (Fig EV2E). The mutated binding motif significantly reduced NICD binding compared to the empty vector (EV) control in CBCs sorted from organoids treated with DMSO (control), JAG1, or DAPT, according to ChIP-qPCR (Fig EV2F), which was consistent with the outcomes of the pull-down assay (Fig 2D). The mutations also significantly decreased Notch1 transcript levels measured by RT–qPCR (Fig EV2G) and NICD levels measured by Western blot (Fig EV2H). Expression levels of Notch signaling components (Notch1, Notch2, Hes1, Hes5) and LGR5 all decreased in CRISPR/Cas9-targeted cells with the mutated binding motif (Fig EV2I). Taken together, the data suggest that, when Notch receptors are activated, the resulting NICD/RBPJk complex bind to the Notch1

---

**Figure 2. Notch1 positive feedback.**

A   Representative image indicating Notch1 expression (red) in intestinal crypt bottoms of tamoxifen-induced Notch1$^{CreER}$ × Rosa26$^{tdTomato}$ mice. Scale bar: 50 μm.
B   Representative images of intestinal crypts showing progeny of Notch1$^{+}$ cells 1 day (left) and 3 days (right) after tamoxifen induction in Notch1$^{CreER}$ × Rosa26$^{tdTomato}$ mice. Dotted lines label the boundary of cells. Scale bar: 20 μm.
C   Luciferase reporter assay of NICD binding sequences. Left: Luciferase reporter vector map with the wild-type and three mutated NICD binding sequences cloned into the enhancer site. Blue represents wild-type sequence, and red represents mutated sequences. Right: Luciferase activity in the four sequences with normalization to a pRL-SV40 control vector. Jag1 or DAPT was added to stimulate or suppress Notch signaling for 48 h. The experiment was performed in replicates (*n* = 4) and presented mean + SEM (*$P \leq 0.05$, **$P \leq 0.01$; Student's *t*-test compares other conditions to WT sequence in normal condition separately).
D   Western blot following DNA pull-down showing NICD-DNA interaction. DNA pull-down was performed using mouse intestine crypt lysates with biotin-labeled oligonucleotide duplex of wild-type or mutated sequences containing the putative NICD/RBPJk binding site, followed by Western blotting to validate NICD binding on the pull-down sequences. Actin was used for input control.
E   Design of gRNAs for CRISPR/Cas9 mutagenesis to target the putative NICD/RBPJk binding motif on mouse Notch1 sequence.
F   Single LGR5-EGFP$^{+}$ CBCs were transfected with either an empty vector (control) or CRISPR/Cas9 gRNAs. Shown are representative brightfield images over 15 days and co-IF images indicating LGR5-EGFP (green) and LYZ (red) expression with DAPI nuclear staining. Scale bar represents 100 μm in low magnification and 25 μm in high magnification images, respectively.
G   Single LGR5-EGFP CBCs were transfected with either an empty vector (control) or CRISPR/Cas9 gRNAs. Left: Colony-forming efficiency measured after 5 days. Quantitative analysis calculated from 1,000 cells/replicate. The experiment was performed in triplicate and presented mean ± SEM (**$P \leq 0.01$; Student's *t*-test). Right: Quantitative comparison of organoid diameters after 15 days. The experiment was performed in triplicate and presented mean ± SEM (**$P \leq 0.01$; Student's *t*-test).
H   Single LGR5-EGFP ISCs were transfected with either an empty vector (control) or CRISPR/Cas9 gRNAs. Ratio of LGR5-EGFP$^{+}$ CBCs/LYZ$^{+}$ Paneth cells as determined by FACS analysis after 15 days. The experiment was performed in triplicate and presented mean ± SEM (**$P \leq 0.01$; Student's *t*-test).
I   Single empty vector control or CRISPR/Cas9-positive feedback knockout (PF KO) LGR5-EGFP$^{+}$ CBCs were transfected with an RBPJk-dsRed reporter construct and grown into organoids, which were subsequently treated with DMSO, DAPT, or JAG1 for 48 h. Left: Representative FACS plots for RBPJk-dsRED and LGR5-EGFP expression indicating a gated double positive fraction for each condition. Right: Mean fluorescence intensity (MFI) of RBPJk-dsRed expression of the entire cell population. The experiment was performed in triplicate and presented mean ± SEM (**$P \leq 0.01$; Student's *t*-test).

Source data are available online for this figure.

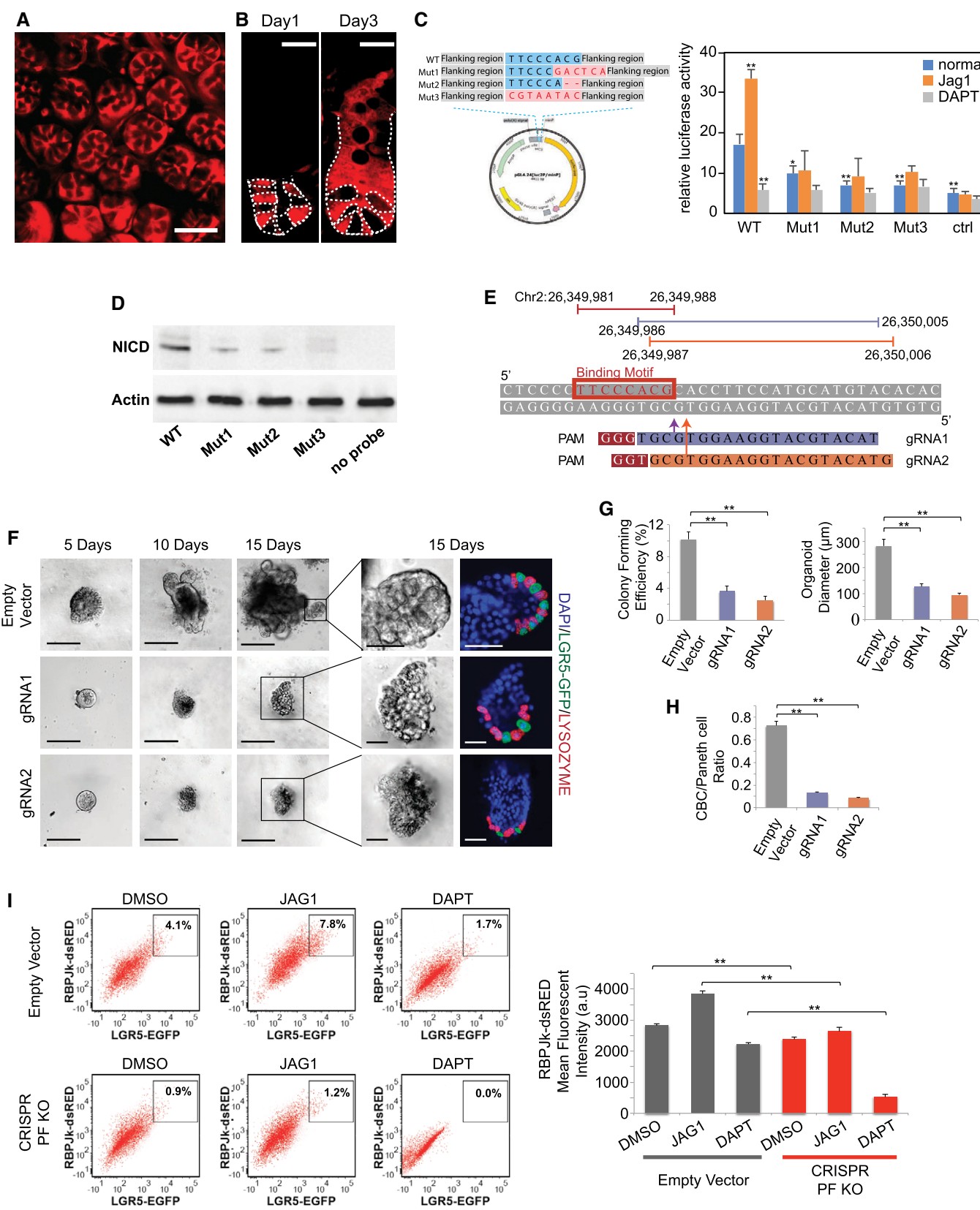

**Figure 2.**

**Figure 3. Notch1 positive feedback is conserved in human colon organoids.**

A   Top: Sequence and chromatogram of NICD binding motif to human Notch1 following ChIP–PCR from EPHB2[high]OLFM4[high] colon stem cells. Bottom: Human Notch1 gene. Red line indicates the location of NICD/RBPJk binding motif on human Notch1.

B   Agarose gel analysis of ChIP–PCR products indicating active binding of NICD to the motif in Notch1 sequence in EPHB2[high]OLFM4[high] colon stem cells treated with DMSO, DAPT, or JAG1.

C   Design of gRNAs for CRISPR/Cas9 mutagenesis to target the putative NICD/RBPJk binding motif on human Notch1.

D   Representative brightfield images of organoids derived from single EPHB2[high]OLFM4[high] colon stem cells transfected with either an empty vector control or CRISPR/Cas9 gRNAs after 7 days (top panel) and 14 days (bottom panel). Scale bar represents 50 μm.

E   Single EPHB2[high]OLFM4[high] human colon stem cells were transfected with either an empty vector control or CRISPR/Cas9 gRNAs. Left: Colony-forming efficiency measured after 7 days. Quantitative analysis calculated from 1,000 cells/replicate. The experiment was performed in triplicate and presented mean ± SEM (**$P \leq 0.01$; Student's $t$-test). Right: Quantitative comparison of organoid diameters after 14 days for each condition. The experiment was performed in triplicate and presented mean ± SEM (**$P \leq 0.01$; Student's $t$-test).

F   Percentage of EPHB2[high]OLFM4[high] stem cells based on FACS analysis for each condition described in (E) after 14 days. The experiment was performed in triplicate and presented mean ± SEM (**$P \leq 0.01$; Student's $t$-test).

G   RT–PCR measurements indicating NOTCH1 expression in EPHB2[high]OLFM4[high] colon stem cells transfected with either an empty vector control or CRISPR/Cas9 gRNAs and subsequently treated with DMSO, DAPT or Matrigel-embedded JAG1. The experiment was performed in triplicate and presented mean ± SEM (**$P \leq 0.01$; Student's $t$-test).

H   Single EPHB2[high]OLFM4[high] colon stem cells were transfected with either an empty vector control or CRISPR/Cas9 gRNAs. Shown is Western blot analysis for NICD expression in sorted EPHB2[high]OLFM4[high] colon stem cells from each condition. Actin was used as a loading control.

I   RT–PCR measurements indicating Notch1/2, Hes1/5, Olfm4, and Lgr5 expression in EPHB2[high]OLFM4[high] colon stem cells for each condition described in (H). The experiment was performed in triplicate and presented mean ± SEM (**$P \leq 0.01$; Student's $t$-test).

gene and enhances its transcription, hence producing more Notch1 receptors and forming a positive feedback loop in intestinal stem cells.

CRISPR mutation of the binding motif (PF KO) reduced colony-forming efficiency and growth rate of intestinal organoids markedly (Fig 2F and G). Furthermore, the mutation significantly reduced the number of CBCs and the ratio of CBC to Paneth cell in the niche (Figs 2F and H, and EV2J). Next, to understand how this positive feedback influences Notch signaling and cell fate, we transfected sorted LGR5-EGFP$^+$ CBCs with an RBPJk-dsRED reporter as an indicator of Notch activity and grew them into organoids, followed by FACS analysis. In PF KO organoids, the Notch[high]/LGR5[high] (dsRed[high]/GFP[high]) CBC population was significantly reduced, responded less to JAG1, and was completed depleted by DAPT compared to the empty vector control (Fig 2I). Therefore, the positive feedback amplifies Notch signaling, and maintains CBC self-renewal and the mosaic niche pattern.

To test whether additional Notch activation can compensate for the Notch1 positive feedback function, CRISPR/Cas9 vectors with specific guide RNAs (gRNAs) were transfected into tamoxifen-inducible NICD-OE intestine cells derived from a LGR5-EGFP-CreERT2 × Rosa26-YFP-NICD strain (Oh *et al*, 2013; Fig EV3). NICD overexpression enhanced colony-forming efficiencies and organoid sizes in both empty vector control and CRISPR mutated organoids (Fig EV3A–C). NICD overexpression largely restored the colony formation efficiencies of mutated organoid to wild-type (empty vector) levels, but not quite their sizes.

**The Notch1 PF motif is conserved in human colon organoids**

Like their mouse counterparts, human intestinal and colon epithelia also contain LGR5$^+$ cells and are highly regenerative. A similar computational analysis of the human genome identified an analogous NICD/RBPJk binding region (TTCCCACG, Chr9: 139,425,108–139,425,115) located on the second intron of the human Notch1 sequence (Fig 3A), which also showed high enrichment of H3K4me1 and H3K27ac enhancer chromatin marks in several

human cell lines (Fig EV4A). We then derived human colon organoids using normal colon tissue in surgically resected specimens from colorectal cancer (CRC) patients (Sato *et al*, 2011b). ChIP-PCR validated NICD binding on the predicted sequence (Fig 3B) in human colon stem cells marked by EPHB2[high]OLFM4[high] expression (Jung *et al*, 2011). Consistent with mouse CBCs, NICD binding to the motif was suppressed by DAPT and enhanced by JAG1 treatment. We then designed CRISPR-Cas9 vectors to mutate the NICD/RBPJk binding motif in human colon stem cells (Figs 3C and EV4B, and Table EV2). ChIP-qPCR validated that the CRISPR/Cas9-mediated mutation reduced NICD binding to the motif in all three conditions (DMSO, JAG1, and DAPT), and prevented JAG1 treatment from increasing NICD binding (Fig EV4C). Suppression of the Notch1 PF by the mutations (PF KO) significantly reduced organoid-forming efficiency, size of organoids, and the percentage of EPHB2[high]OLFM4[high] colon stem cells compared to empty vector control (Figs 3D–F and EV4D). The epithelial cell identity of the EPHB2[high]OLFM4[high] cells was validated by their EpCAM expression (Fig EV4D). Without the signal-amplifying Notch1 PF, colon stem cells had lower Notch1 transcript levels (Fig 3G). Mutated colon stem cells also had lower expression levels of NICD, other Notch signaling components, and human colon stem cell markers (LGR5 and OLFM4) (Fig 3H and I). In summary, the Notch1 PF promotes Notch signaling and self-renewal in human colon stem cells.

**Notch1 PF enhances robustness in dynamic stem cell niche models**

The niche maintains a mosaic Notch[high] (CBC) and Notch[low] (Paneth) cell pattern for proper homeostasis. It seems that lateral inhibition (LI) alone should be sufficient for generating such patterns as it did for other developmental patterning (Collier *et al*, 1996). The positive feedback increases Notch signaling levels in CBCs, but, from a "control" perspective, LI could hypothetically achieve the same objective by simply increasing Notch1 transcription/translation rates. To understand whether the addition of the Notch positive feedback to LI (PFLI) offer any regulatory advantages

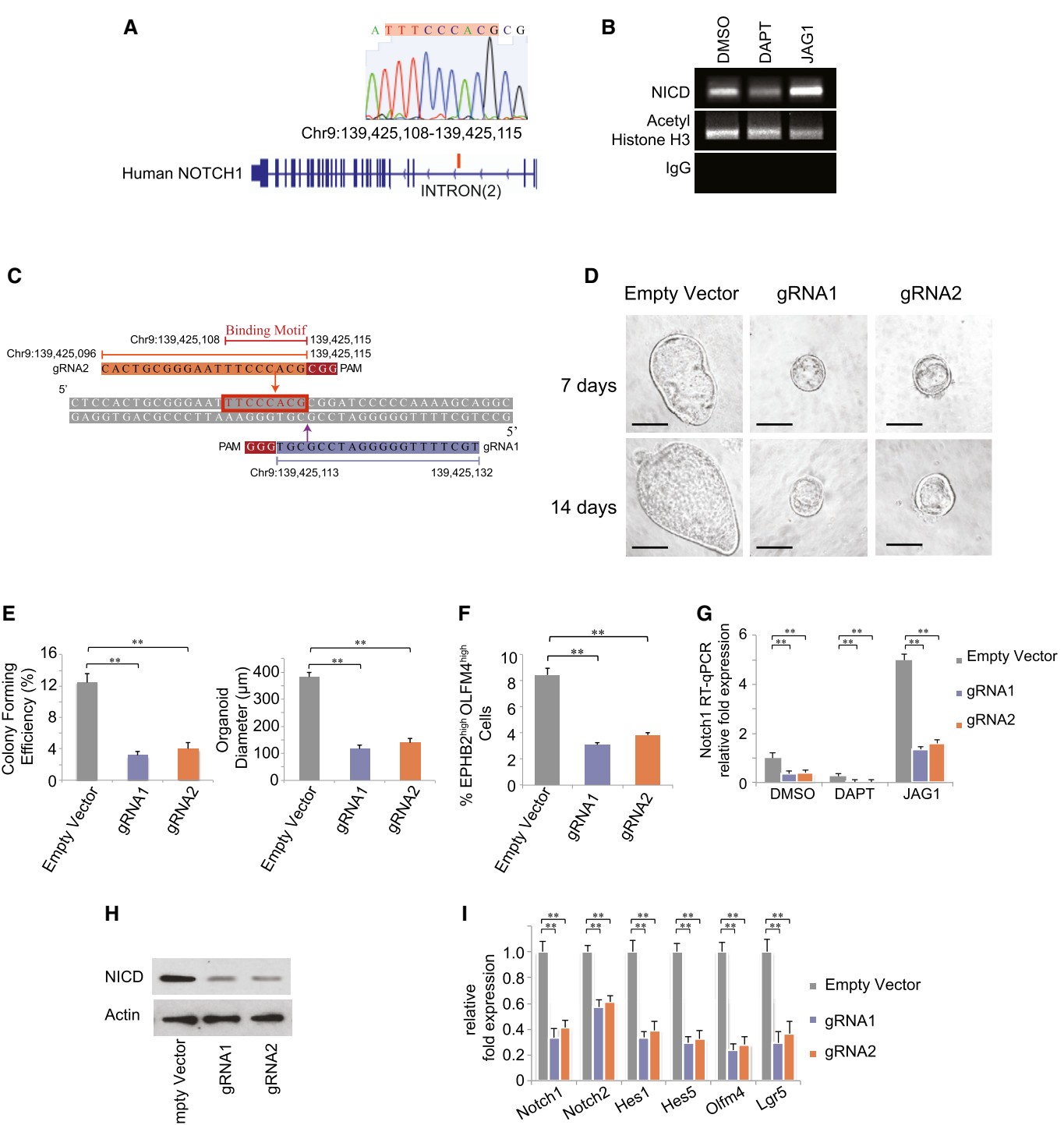

**Figure 3.**

that cannot be achieved by the LI motif alone, we constructed single-, pair-, and multicell mathematical models to analyze LI with and without PF (Materials and Methods, Table EV3).

We first analyzed Notch response to external ligands in a single cell as an input–output function. In LI, external ligands activate receptors and Notch signaling, which in turn suppresses internal ligand expression. Hence, increasing levels of external ligands leads to a monotonic decrease of internal ligands (Figs 4A and EV5A). On the other hand, PFLI causes bifurcation and generates a more switch-like response with hysteresis (Figs 4A and EV5B).

Pair-cell analysis suggests that both LI and PFLI could achieve intercellular bistability, with two neighboring cells settling in opposite (Notch[high] vs. Notch[low]) states. Nevertheless, PFLI is much more robust in generating bistability than LI alone and is less

dependent on cooperativity (Hill coefficient) of the reactions (Figs 4B and EV5C and D).

Next, we used the maximum Lyapunov exponent (MLE) to analyze the stability of patterning in multicellular systems (Sprinzak *et al*, 2011). PFLI is able to maintain stable patterns over a much wider parameter range than LI, especially when cooperativity of reaction is low, suggesting that PFLI is a more robust patterning mechanism (Fig 4C). We then scaled up the pair-cell model to a multicellular model with stationary cells surrounding each other to explore Notch patterning dynamics. In steady state analysis, both LI and PFLI can generate stable mosaic patterns with varying levels of Notch signaling and ligands (Fig EV6A). However, dynamic simulations from an initial homogeneous state suggest that PFLI reaches the steady state pattern much faster than LI by speeding up divergence of individual cell signaling states (Fig 4D). Taken together, the analyses suggest that the positive feedback increases robustness, stability, and speed of Notch-mediated patterning.

These properties are not necessarily important for relatively stationary patterns with low cellular turnovers. However, they could be important during regeneration when cell divisions and migration perturb the pattern. We therefore constructed a multiscale, agent-based stochastic model using CompuCell3D (Swat *et al*, 2012). The model takes into consideration the three-dimensional structure, cell growth, division, migration, and cell–cell physical contact (Fig 4E, see Materials and Methods). Notch signaling is only modeled at the base of the crypt, while cells above the niche are simply pushed upwards with no specific assumptions made about their properties. This model does not attempt to capture every aspect of the crypt or the exact movements or lineages of individual cells, which involve many signaling pathways and types of cell–cell interaction. Rather, it is solely designed to test how cell division and migration would affect PFLI- vs. LI-mediated Notch patterning.

As expected, PFLI generates bimodality in cells with regard to Notch signaling (NICD) levels (Fig EV6B). When the strength of the PF is reduced, the ratio of NICD$^{high}$ to NICD$^{low}$ cells as well as NICD levels in the NICD$^{high}$ population decrease (Fig EV6B), consistent with our experimental observation that CRISPR/Cas9-mutated PF KO organoids have lower CBC/Paneth cell ratio, and those CBCs have weaker signals (Figs 2F and G, and EV2G–J).

However, can the Notch signaling pattern be maintained by LI alone if we simply change its parameters to increase Notch signaling levels? In other words, is the PF's role limited to maximizing Notch signaling levels, or is PFLI an inherently different control scheme from LI? To address this hypothetical question, we readjusted the maximum Notch transcription rate in the LI model, so that LI and PFLI have equivalent Notch signaling levels. Indeed, both LI and PFLI can generate mosaic Notch signaling patterns when cell proliferation rate is slow (Fig 4E). We then gradually increased the proliferation rate, which leads to increased rates of cell division, migration, and anoikis (Fig EV6C). PFLI is still able to maintain bimodality and binary patterns, whereas LI starts to show less bimodality (more cells with intermediate Notch levels) and more blurred pattern (Fig 4E). Therefore, PFLI is an inherently more robust control motif than LI when the pattern regulation is dynamic and has to cope with increasing rate of cell division, turnover, and movement.

## Discussion

The intestinal stem cell niche controls regeneration and homeostasis of the tissue. Here, we report a direct positive feedback, in which NICD cleaved from activated receptors directly bind to the second intron of Notch1 gene to enhance its expression. This positive feedback is active in mouse intestinal and human colon epithelial cells, and its silencing by CRISPR/Cas9 mutation reduced CBC/Paneth cell ratio and limited self-renewal. Dynamical systems analysis and multiscale stochastic simulation further suggest that the PFLI architecture has an inherent advantage over LI with regard to robustness if the signaling pattern is dynamic.

Biological systems such as the stem cell niche are usually robust (Savageau, 1971; Barkai & Leibler, 1997; Alon *et al*, 1999; Stelling *et al*, 2004; Kitano, 2007; Shen *et al*, 2008). They work most of the time, capable of accommodating different conditions and recovering from mistakes and damages. Control theory would predict that they rely on additional mechanisms such as feedback to enhance their regulation. In fact, recent reports showed that Wnt protein works as short-range signals in the intestinal stem cell niche and there is a positive feedback involving Ascl2 and Wnt signaling in intestinal stem cells (Schuijers *et al*, 2015; Farin *et al*, 2016). From an individual cell perspective, positive feedback loops translate gradients into binary decision states, as shown by the single-cell analysis. From a multicellular signaling pattern perspective, such positive feedbacks

**Figure 4. Computational analysis of Notch patterning with lateral inhibition and positive feedback.**

A   Dynamic analysis of the single-cell Notch signaling model. Internal Dll ($D_i$) vs. external Dll ($D_{ext}$) protein levels are plotted. Lateral inhibition (LI) exhibits monostable behavior (top panel) while Notch positive feedback + LI (PFLI) exhibits bifurcation (bottom panel) in response to external Notch activation. The schematic illustrations on the left of each panel show LI and PFLI in single cells with external Dll ($D_{ext}$) inducing Notch signaling activation and following regulation on Notch ($N_i$) Dll ($D_i$).

B   Phase portraits of the pair-cell Notch signaling model. LI (top panel) requires higher cooperativity (Hill coefficient, h) than PFLI (bottom panel) to generate bistability. Lines: nullclines; solid dots: stable steady states; hollow dots: unstable steady states. $N_i$, $N_j$ and $D_i$, $D_j$ refer to Notch and Dll levels in cells i and j, respectively.

C   Multicell maximum Lyapunov exponents (MLE) analysis of LI-only (top) or PFLI (bottom) circuits spanning parameters of production rates of Notch and Dll with varying degrees of cooperativity (H). Yellow regions (positive MLE values) represent states with patterning, and green regions (negative MLE values) represent states without patterning. PFLI generates patterns over a broader parameter range than LI.

D   Multicellular simulation of Notch signaling models showing DLL levels from initially homogeneous unstable steady states to heterogeneous stable steady states (top panel). Middle-left panel: Representative simulation of LI. Middle-right panel: Representative simulation of PFLI. Bottom panel shows the change rates of DLLs levels during patterning with varying relative strength of positive feedback. Red: Signaling dynamics of cells reaching high steady states. Blue: Signaling dynamics of cells reaching low steady states. Gray: Patterning transition period from homogeneous steady states to heterogeneous steady state.

E   Analysis of a stochastic crypt model integrated with Notch signaling simulation. Left: Structure of the crypt model. Right: Representative violin plots indicating NICD level dynamics at crypt bottom with varying turnover rates. Shown also are corresponding representative simulated patterns of NICD levels in crypt bottoms. PFLI shows stronger NICD bimodality and binary patterns than LI when turnover rates become higher.

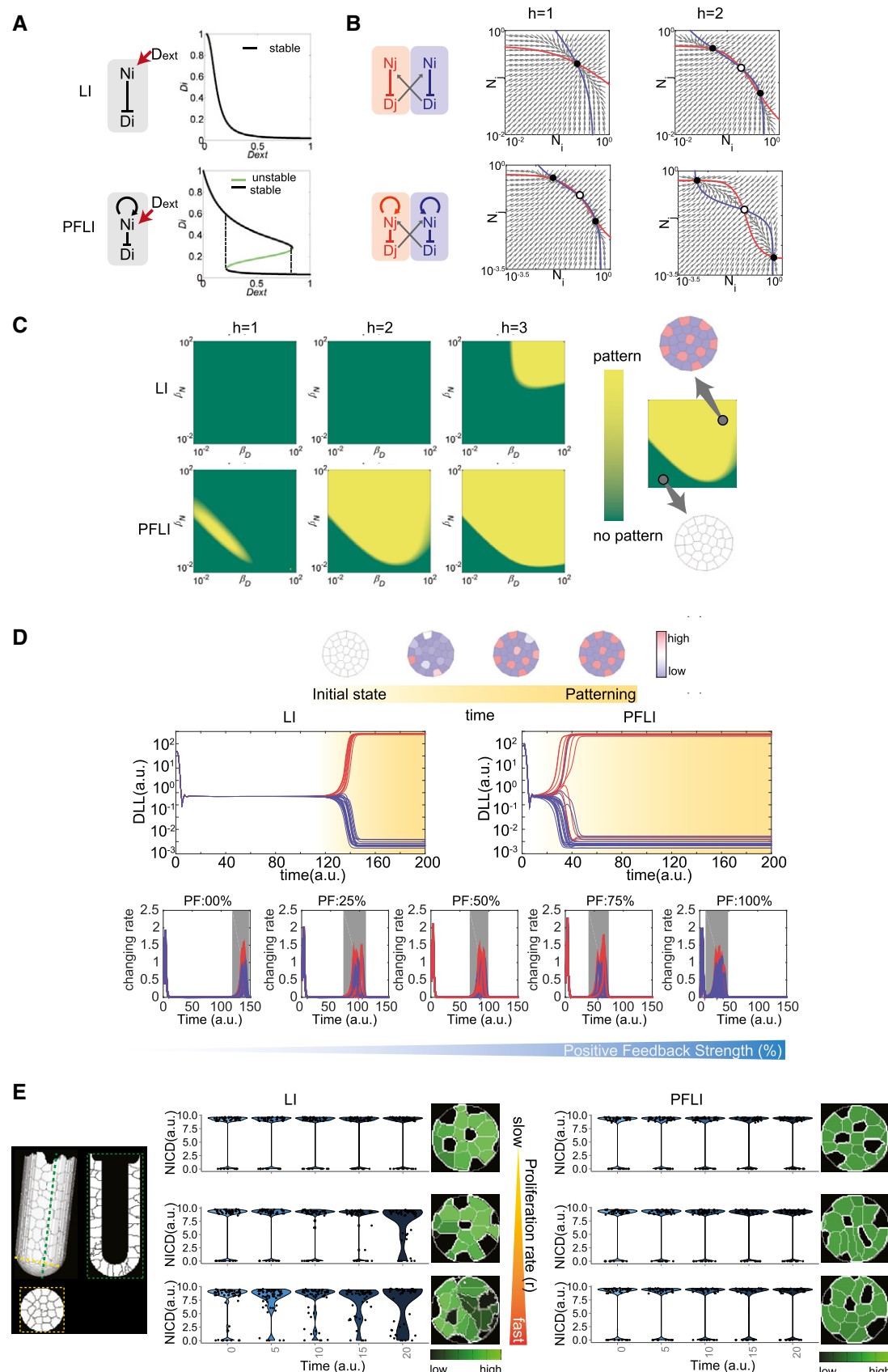

**Figure 4.**

**Table 1.  Table of antibodies.**

| Primary antibody | Supplier | Catalog number | Dilution[a] |
|---|---|---|---|
| anti-ATOH1 | Abcam | ab137534 | 1:1,000 (WB) |
| anti-β-actin | Abcam | ab6276 | 1:4,000 (WB) |
| Anti-CD24 (APC) | Abcam | ab51535 | 1:500 (FC) |
| Anti-DLL1 | Abcam | ab85346 | 1:500 (WB) |
| Anti-DLL4 | Abcam | ab7280 | 1:1,000 (WB) |
| Anti-EPHB2 | R&D Systems | AF467 | 1:1,000 (FC) |
| Anti-EpCAM (FITC) | Abcam | ab8666 | 1:500 (FC) |
| Anti-GFP | Abcam | ab5450 | 1:200 (IF) |
| Anti-HES1 | Abcam | ab108937 | 1:1,000 (WB) |
| Anti-HES5 | Santa Cruz Biotechnology | sc-25395 | 1:500 (WB) |
| Anti-JAG1 | Santa Cruz Biotechnology | sc-6011 | 1:500 (WB) |
| Anti-lysozyme | Abcam | ab108508 | 1:100 (IF) |
| Anti-human NICD | R&D Systems | AF3647 | 1:200 (ChIP) |
| Anti-mouse NICD | Cell Signaling | 4147 | 1:200 (ChIP) |
| Anti-NOTCH1 | Santa Cruz Biotechnology | sc-9170 | 1:1,000 (WB) |
| Anti-NOTCH2 | Santa Cruz Biotechnology | sc-32346 | 1:1,000 (WB) |
| Anti-OLFM4 | Sino Biological Inc. | 11639-MM12-50 | 1:1,000 (FC) |

[a]Application: IF, immunofluorescence; WB, Western blotting; FC, flow cytometry.

can enhance the robustness of the patterns, making sure that signaling states are sustained and not easily disrupted by the change of surrounding cells. Therefore, the Notch1 and Ascl2/Wnt positive feedback loops (and potentially many others) may reflect a general regulatory scheme for such systems. Other mechanisms such as cis-inhibition between Notch receptors and ligands (Sprinzak *et al*, 2010) may further enhance robustness. An approach that combines experimental methods and computational models (Collier *et al*, 1996; Johnston *et al*, 2007; Buske *et al*, 2011) may help us not only identify such mechanisms but also understand their contribution to regulation of the intestinal stem cell niche.

In this study, recombinant JAG1 ligands were embedded in Matrigel to activate Notch signaling in intestinal organoid cells as an alternative to EDTA-mediated Notch signaling activation. Notch ligands are thought to activate receptors via mechanical pulling—hence hybridization of ligands to a surface seems to be required (Wang & Ha, 2013; Gordon *et al*, 2015). However, ligand hybridization to a 2D surface does not work for 3D cell culture. On the other hand, recombinant ligands such as Dll1, Dll4 (Hicks *et al*, 2002; Lefort *et al*, 2006; Mohtashami *et al*, 2010; Xu *et al*, 2012; Castel *et al*, 2013), and JAG1 (Sato *et al*, 2009; Van Landeghem *et al*, 2012; VanDussen *et al*, 2012; Yamamura *et al*, 2014; Mahapatro *et al*, 2016; Srinivasan *et al*, 2016) have been reported to activate

Notch signaling in various cell types, including intestinal organoids and tumor spheroids. One possibility is that ligands embedded in the Matrigel provide the necessary pulling force to activate receptors, although the exact mechanism is still not clear.

## Materials and Methods

### Mouse strains

LGR5-EGFP mice on a mixed 129/C57BL/6 background and Rosa26-CAG-LSL-tdTomato-WPRE mice on a mixed 129/C57BL/6 background were purchased from The Jackson laboratory. Notch1-CreERT2 knock-in (KI) mice, Notch2-CreERT2 KI mice, and Rosa26-NICD-IRES-YFP KI mice were a generous give from Dr. Spyros Artavanis-Tsakona's laboratory at Harvard University. Subsequently, we generated an inducible Notch1 reporter mouse strain (Notch1-CreERT2 KI × Rosa26-CAG-LSL-tdTomato-WPRE) and an inducible Notch2 reporter mouse strain (Notch2-CreERT2 KI × Rosa26-CAG-LSL-tdTomato-WPRE). We also generated a LGR5-EGFP-CreERT2 × Rosa26-NICD-IRES-YFP KI mouse strain for inducible NICD overexpression (NICD-OE). Genotyping was performed using the following PCR primer pairs for Notch1 (forward: ATAGGAACTTCAAAATGTCGCG; reverse: CACACTTC CAGCGTCTTTGG), Notch2 (forward: ATAGGAACTTCAAAATGT CGCG; reverse: CCCAACGGTGCCAAAAGAGC), and NICD (forward: CTTCACACCCCTCATG ATTGC; reverse: GCAATCGGTCCATGTGA TCC). The thermocyling profile used for PCR amplification is described as follows: 95°C (5 min)/[95°C (30 s), 60°C (30 s), 72°C (60 s)] for 35 cycles/72°C (10 min). Notch1 and Notch2 reporter mice were induced with 75 mg/kg tamoxifen by i.p. injection. The LGR5-EGFP-CreERT2 × Rosa26-NICD-IRES-YFP KI mouse strain was treated daily with 75 mg/kg tamoxifen (i.p. injection) for eight consecutive days to induce Cre enzyme activity and NICD-OE phenotype. All animal experiments were approved by the Cornell Center for Animal Resources and Education (CARE).

### Murine intestinal crypt isolation and organoid culture

Eight-week-old LGR5-EGFP or LGR5-EGFP/NICE-OE mice were sacrificed to establish intestinal organoid culture. Briefly, small intestines were harvested, flushed with $Ca2^+/Mg^{2+}$-free PBS to remove debris, and opened up longitudinally to expose luminal surface. A glass coverslip was then gently applied to scrape off villi, and the tissue was cut into 2–3 mm fragments. Intestinal tissues were then washed again with cold PBS and incubated with 2.0 mM EDTA for 45 min on a rocking platform at 4°C. EDTA solution was then decanted without disturbing settled intestinal fragments and replaced with cold PBS. In order to release intestinal crypts in solution, a 10-ml pipette was used to vigorously agitate tissues. The supernatant was collected, and this process was repeated several times to harvest multiple fractions. The crypt fractions were then centrifuged at 84 *g* for 5 min. Based on microscopic examination, the appropriate enriched crypt fractions were pooled and centrifuged again to obtain a crypt-containing pellet. Advanced DMEM/F12 (Life Technologies) containing Glutamax (Life Technologies) was used to resuspend the cell pellet and subsequently a 40-μm filter was used to purify crypts. Next, single-cell dissociation was

achieved by incubating purified crypt solution at 37°C with 0.8 KU/ml DNase (Sigma), 10 μM ROCK pathway inhibitor, Y-27632 (Sigma), and 1 mg/ml trypsin-EDTA (Invitrogen) for 30 min. Single cells were then passed again though a 40-μm filter and resuspended in cold PBS with 0.5% BSA for FACS analysis to collect LGR5-EGFP$^+$ intestinal stem cells (ISCs), which are also called crypt base columnar (CBC) cells.

Single LGR5-EGFP$^+$ CBCs were suspended in Matrigel (BD Biosciences) at a concentration of 1,000 cells or crypts/ml, and 50 μl Matrigel drops were seeded per well on pre-warmed 24-well plates. Matrigel polymerization occurred at 37°C for 10 min and was followed by the addition of complete media to each well. ISC media included the following: Advanced DMEM/F12 supplemented with Glutamax, 10 mM HEPES (Life Technologies), N2 (Life Technologies), B27 without vitamin A (Life Technologies), and 1 μM N-acetyl-cysteine (Sigma). Growth factors were freshly prepared each passage in an ISC media solution containing 50 ng/ml EGF (Life Technologies), 100 ng/ml Noggin (Peprotech), and 10% R-spondin1-conditioned media (generated in house). The addition of growth factors occurred every 2 days, and the media were fully replaced every 4 days. Organoids were passaged once per week at a ratio of 1:4 by removing organoids from Matrigel with ice-cold PBS. Next, organoids were incubated on ice for 10 min followed by mechanical disruption, centrifugation, and resuspension in fresh Matrigel.

For *in vitro* studies, organoids derived from single LGR5-EGFP ISCs were treated with one of the following: DMSO or 10 μM DAPT (EMD Millipore) added to the media for 48 h (Sikandar *et al*, 2010), or 1uM JAG-1 (AnaSpec) embedded in Matrigel for 48 h (Takeda *et al*, 2011). EDTA was added to ISC media (for a final concentration of 0.5 mM EDTA) to treat organoids for 4 h. Subsequently, organoids were harvested and analyzed by FACS to isolate LGR5-EGFP cells and Paneth cells for RT–PCR or protein analysis. FACS was conducted using a Beckman Coulter flow cytometer with a 40-μm filter. Data analysis was performed using FlowJo software to gate populations according to 7-AAD viability, and forward and side scattering. Cutoff thresholds were provided by unstained ISCs and single stained ISCs when using multiple fluorochromes in order to achieve appropriate compensation.

### CRISPR/Cas9 genomic editing

The procedure for CRISPR/Cas9-mediated transfection in mouse ISC organoids has been previously described (Schwank *et al*, 2013). Briefly, guide RNA (gRNA) sequences were designed by Optimized CRISPR Design tool (http://crispr.mit.edu/), and CRISPR/Cas9 plasmids including gRNA sequences were purchased from GenScript. For murine experiments, gRNAs targeting the NICD binding motif on Notch1 included the following: gRNA1: (TACATGCAT GGAAGGTGCGT) or gRNA2: (GTACATGCATGGAAGGTGCG) and were cloned into a pGS-gRNA-Cas9-Puro vector backbone. A pGS-CAS9-PURO only vector (no gRNA) was used as a control. Single sorted LGR5-EGFP$^+$ ISCs were transfected using Lipofectamine-2000 (Life Technologies) according to the manufacturer's instructions. Briefly, 4uL Lipofectamine-2000 (diluted in 50 μl Opti-MEM) and 2 μg of CRISPR/Cas9 plasmids (diluted in 50 μl Opti-MEM) were mixed 1:1: and incubated for 5 min at room temperature. Lipofectamine/DNA complexes were then added to single LGR5-EGFP$^+$

ISCs (50 μl/well) in a 24-well plate, which was subsequently centrifuged for 1 h and incubated at 37°C for 4 h. ISCs were then resuspended in Matrigel and overlaid with ISC media (as prepared above) and supplemented with Y-27632 for 48 h. Next, transfected ISCs were selected in media (without R-spondin) containing 300 ng/μl puromycin for 72 h. Selection media were then replaced with ISC media, and organoids were monitored for 15 days followed by FACS analysis or co-immunofluorescence. Individual CRISPR/Cas9-mutated organoid clones were also harvested and lysed for DNA extraction using a QIAmp DNA Mini kit (Qiagen: 51304) according to the manufacturer's instructions. Subsequently, the NICD binding site on mouse Notch1 was amplified by PCR using the following primers (forward: AGAAGAGAAGACAGGAGAAGGA and reverse: GAAGCCACTGACTTTCCTAGAG) and analyzed by Sanger sequencing to visualize mutations. CRISPR/Cas9-mutated organoids derived from single transfected LGR5-EGFP ISCs were also treated for 48 h with DAPT or JAG-1 (as described earlier) before harvesting cells for FACS to isolate LGR5-EGFP$^+$ cells for RT–PCR analysis.

In order to study Notch signaling dynamics, a RBPJk-dsRed reporter on a pGA981-6 vector backbone (Addgene #47683) was transfected into single wild-type or CRISPR/Cas9-mutated sorted LGR5-EGFP ISCs using Lipofectamine-2000 according to the protocol described above. ISCs were then treated for 48 h with 10 μM DAPT or 1 μM JAG-1 and analyzed by microscopy and flow cytometry for LGR5-EGFP and RBPJk-dsRed expression.

### Isolation of single cells from human colonic tissue

Approval for this research protocol was obtained from IRB committees at Weill Cornell Medical College and NY Presbyterian Hospital. Patients undergoing colorectal surgery provided written informed consent for use of human tissues. Material was derived from proximal colonic tissue during surgical biopsies. The procedure for isolation of colonic stem cells and organoid culture is previously described (Jung *et al*, 2011). Briefly, colonic specimens were collected and incubated in Advanced DMEM/F12 supplemented with gentamycin (Life Technologies) and fungizone (Life Technologies). Extraneous muscular and sub-mucosal layers were removed from colonic mucosa. The tissue was cut into 1 cm fragments and incubated with 8 mM EDTA for 1 h on a rocking platform at 37°C followed by a 45-min incubation at 4°C. The supernatant was replaced with Advanced DMEM/F12 supplemented with Glutamax, HEPES, and 5% FBS. Vigorous shaking released crypts, which were collected in several fractions. Crypt fractions were then centrifuged (37 *g*, 5 min) and visualized by microscopy to determine which enriched fractions to combine. Subsequently, pooled crypt fractions were centrifuged and resuspended in Advanced DMEM/F12 supplemented with Glutamax, HEPES, N-2, B-27 without vitamin A, 1 mM N-acetyl-L-cysteine, nicotinamide (Sigma), 10 μM Y-27632, 2.5 μM PGE2 (Sigma), 0.5 mg/ml Dispase (BD Biosciences), and 0.8 KU/ml DNase I. Cells were then incubated for 15 min at 37°C followed by mechanical disruption and passage of cell solution through 40-μm filter to obtain a single-cell suspension.

### Human colon organoid culture

Single human colon cells were stained with EPHB2 (conjugated to PE), OLFM4 (conjugated to APC), EpCAM-FITC, and 7-AAD

according to standard protocols and were suspended in cold PBS with 0.5% BSA for FACS analysis. FlowJo software was used to gate populations according to 7-AAD viability, and forward and side scattering. Cutoff thresholds were provided by unstained ISCs and single stained ISCs when using multiple fluorochromes in order to achieve appropriate compensation. EPHB2$^{high}$OLFM4$^{high}$ colon stem cells were harvested, and subjected to lipotransfection of CRISPR/Cas9 plasmids using Lipofectamine-2000 in a similar method as described earlier for mouse ISCs. CRISPR/Cas9 gRNAs targeting the NICD binding motif of human Notch1 (cloned into a pGS-gRNA-Cas9-Puro vector backbone) were designed and ordered from GenScript with the following inserted gRNA sequences: (gRNA1: TGCTTTT GGGGGATCCGCGT, gRNA2: CACTGCGGGAATTTCCCACG). A pGS-CAS9-PURO only vector (no gRNA) was used as a control. Transfected human colon stem cells were selected in medium lacking WNT-3A, and R-spondin1 and containing Y-27632 and 300 ng/μl puromycin for 48 h.

Subsequently, transfected cells were suspended in Matrigel, and overlaid with human colon stem cell medium containing Advanced DMEM/F12 supplemented with Glutamax, HEPES, N-2, B-27 without vitamin A, 1 mM N-acetyl-L-cysteine, nicotinamide, PGE2, Y-27632, human Noggin (Peprotech), human EGF (Life Technologies), gastrin (Sigma), TGF-β type I receptor inhibitor A83-01 (Tocris), P38 inhibitor SB202190 (Sigma-Aldrich), WNT3A-conditioned media (generated in house), and R-spondin1-conditioned medium (generated in house) (Jung *et al*, 2011). For organoid culture, full medium was replaced every 2 days. Transfected organoids were monitored for 14 days and then harvested and analyzed by FACS to isolate EPHB2$^{high}$OLFM4$^{high}$ colon stem cells for RT–PCR and protein analysis. Individual CRISPR/Cas9-mutated organoid clones were also harvested and lysed for DNA extraction using a QIAmp DNA Mini kit (Qiagen: 51304) according to the manufacturer's instructions. Subsequently, the NICD binding site on human Notch1 was amplified by PCR and analyzed by Sanger sequencing to visualize mutations.

### Quantitative RT–PCR and protein analysis

Total RNA from mouse ISCs or human colon stem cells was extracted using a Qiagen RNeasy Plus kit. Subsequently, isolated RNA was reverse transcribed to cDNA using ABI Taqman Reverse Transcription kit (Applied Biosystems). ABI Taqman Master mix and ABI Prism HT7900 were used to run quantitative real-time PCR. Taqman primers (ABI) purchased from Life Technologies were used for the following mouse genes: Notch1 (Product ID: Mm00627185_m1), Notch2 (Product ID: Mm00803077_m1), Hes1 (Product ID: Mm01342805_m1), Hes5 (Product ID: Mm00439311_g1), Dll1 (Product ID: Mm01279269_m1), Dll4 (Product ID: Mm00444619_m1), Jag1 (Product ID: Mm00496902_m1), Atoh1 (Product ID: Mm00476035_s1), Lgr5 (Product ID: Mm00438890_m1). Human Taqman primers purchased from Life Technologies include: Notch1 (Product ID: Hs01062014_m1), Notch2 (Product ID: Hs01050702_m1), Hes1 (Product ID: Hs00172878_m1), Hes5 (Product ID: Hs01387463_g1), Lgr5 (Product ID: Hs00969422_m1), and Olfm4 (Product ID: Hs00197437_m1). RT–PCR analysis represents the average of three independent experiments normalized to GAPDH expression. Error bars designate SEM. Protein extraction from mouse ISCs or human

colon stem cells and Western blotting were performed as previously described. β-actin was used as a control for normalization (Pan *et al*, 2008). Antibodies used for Western blotting are listed in Table 1.

### ChIP-PCR

Mouse intestinal and human colonic organoids were harvested, and ChIP-PCR was performed according to manufacturer's instructions (EMD Millipore: 17-408). Briefly, normal rabbit IgG was used as a negative control while rabbit anti-acetyl histone H3 was used as a positive control for immunoprecipitation (IP). Subsequently, primer pairs specific for human or mouse GAPDH sequences were for positive PCR controls. Following IP using anti-mouse NICD, PCR primers (forward: AGATGAAGGTGGAGCATGTG, reverse: TTTTCC CACGGCCTAGAAG) were used for amplification of Notch1. Similarly, for ChIP assays involving anti-human NICD, PCR primers (forward: ACTAGGTGTCACCAAAGTGC, reverse: CATGACCATCTTG GCCTCTC) were used to amplify Notch1. Sanger sequencing was used to validate NICD binding motif on Notch1 for PCR products. Subsequently, ChIP-qPCR analyses were performed according to the manufacturer's instructions (Active Motif: 53029). Antibodies used for ChIP are listed in Table 1.

### Immunofluorescence

Intestinal tissues from tamoxifen-induced Notch1 (tdTomato) reporter mice were harvested at various time points, fixed with 4% PFA, snap-frozen in O.C.T., cryo-sectioned, and visualized on a Zeiss LSM 510 laser scanning confocal microscope. DAPI was used as a nuclear counterstain. For *in vitro* imaging, wild-type or CRISPR/Cas9-mutated intestinal organoids derived from LGR5-EGFP mice were embedded in Matrigel on glass chamber slides. Cells were fixed for 15 min at room temperature using 4% PFA and rinsed three times with PBS. 0.2% Triton X-100 was used for permeabilization of cell membranes. Next, cells were incubated in a serum-free blocking solution (Dako) for 30 min. For co-immunofluorescence staining, an antibody diluent solution (Dako) was used to prepare primary and secondary antibodies. Primary antibodies were added overnight at room temperature followed by application of Alexafluor 488/555 secondary antibodies for 1 h. Organoids were visualized using lysozyme (LYZ) and LGR5 (detected by GFP) expression. DAPI (Life Technologies) was as a nuclear counterstain on a Zeiss LSM 510 laser scanning confocal microscope using an Apo 40× NA 1.40 oil objective. Antibodies used for immunofluorescence are listed in Table 1.

### Luciferase assay

The wild-type (WT) enhancer sequence and three mutated sequences were PCR amplified (WT:CTGTCAACCTTGCTTCCTCCC C**ttcccac**g**c**GCACCTTCCATGCATGTACACAC, Mut1: CTGTCAACCTTG CTTCCTCCCC**ttcccgactca**CTTCCATGCATGTACACAC, Mut2: CTGT CAACCTTGCTTCCTCCCC**ttccca**CACCTTCCATGCATGTACACAC, and Mut3: CTGTCAACCTTGCTTCCTCCCC**cgtaatac**CACCTTCCATGCAT GTACACAC) and cloned into a pGL4.24 firefly luciferase reporter plasmid (Promega). These luciferase reporter vectors and *Renilla* luciferase vector (pRL-SV40, Promega) were co-transfected into

mouse intestine cells using Lipofectamine 3000 (Life Technologies) according to the manufacturer's instructions. Cell lysates were collected, and luciferase samples were prepared using the Luc-Pair Duo-Luciferase Assay kit (Genecopoeia) in 48 h after transfection. Firefly luciferase activities were measured using an FLUOstar optima plate reader (BMG Labtech), and firefly luciferase activity was normalized to *Renilla* luciferase activity.

**Biotinylated nucleotide pull-down assay**

Oligonucleotides of the wild-type and three mutated sequences (same as in the luciferase assay) were labeled using a biotin labeling kit (Pierce) and annealed for pull-down assay. Mouse intestine crypt cell lysates were freshly prepared using RIPA buffer (Millipore) with proteinase inhibitor (Roche). After precleared using Dynabeads M-270 streptavidin (Invitrogen), the cell lysates were diluted in binding buffer and incubated with the biotinylated DNA duplex for 2 h at 4°C. Dynabeads M-270 streptavidin were then added into the mixture and incubated for 1 h at 4°C. After washing, the DNA-binding protein complexes were released from the Dynabeads. The retrieved proteins were collected for Western blot validation.

**Statistical analysis**

The data are displayed as mean $\pm$ SEM. Statistical comparisons between two groups were made using Student's *t*-test. $P < 0.05$ was used to establish statistical significance.

**ODEs models of Notch signaling circuits**

The mathematical model of Notch signaling includes three types of regulations: (i) trans-activation of Notch receptor by external ligand (TA), (ii) lateral Inhibition (LI), and (iii) positive feedback (PF). Transcriptional activation is modeled by Hill function $\sigma(x, k, p) = (x^p/(k^p + x^p))$, and transcriptional suppression is modeled by Hill function $\delta(x, k, h) = (k^h/(k^h + x^h))$, where $x$ refers to the regulator, $k$ refers to the saturation coefficient, and $h$ is the Hill coefficient. Below are the ODE equations:

$$\dot{Notch}_{mRNAi} = \beta_{n0} + \beta_n \cdot \sigma(k_t \cdot NOTCH_i \langle DLL_j \rangle, k_p, p) - \alpha_n \cdot Notch_{mRNA_i}$$
$$\dot{NOTCH}_i = \beta_N \cdot Notch_{mRNA_i} - \alpha_N \cdot NOTCH_i - k_t \cdot NOTCH_i \langle DLL_j \rangle$$
$$\dot{Dll}_{mRNA_i} = \beta_{d0} + \beta_d \cdot \delta(k_t \cdot NOTCH_i \langle DLL_j \rangle, k_d, h) - \alpha_d \cdot Dll_{mRNA_i}$$
$$\dot{DLL}_i = \beta_D \cdot Dll_{mRNA_i} - \alpha_D DLL_i - k_t \cdot \langle NOTCH_j \rangle DLL_i$$
$$\dot{R}_i = k_t NOTCH_i \langle DLL_j \rangle - \alpha_R R_i$$

where $Notch_{mRNA}$, $NOTCH$, $Dll_{mRNA}$, $DLL$, and $R$ refer to the expression level of Notch mRNA, NOTCH receptor, Dll mRNA, DLL ligand, and cleaved NICD (activated Notch signaling), respectively. The annotation $i$ and annotation $j$ refer to cell $j$ adjacent to cell $i$. $\beta_s$ denote the synthesis rates (transcription rates for mRNAs and translation rates for protein), while $\alpha_s$ denote the degradation rates. $\langle X_j \rangle_i$ is the average expression of $X$ from the neighboring $j$ cells of cell $i$. $k_t$ is the reaction rate of trans-activation. $\beta_{n0}$ and $\beta_{d0}$ are the basal transcriptional rates of $Notch_{mRNA}$ and $Dll_{mRNA}$. By changing the ratios of $\beta_n/(\beta_{n0} + \beta_n)$ and $\beta_d/(\beta_{d0} + \beta_d)$, we can adjust the regulatory strength of LI and PF.

For simplicity, we transform the equations into dimensionless equations with dimensionless parameters:

$$\tau \equiv t_0 t, Nm \equiv \frac{Notch_{mRNA}}{Nm_0}, N \equiv \frac{NOTCH}{N_0}, Dm \equiv \frac{Dll_{mRNA}}{Dm_0},$$
$$D \equiv \frac{DLL}{D_0}, R \equiv \frac{R}{R_0}, N_0 = D_0 = R_0 \equiv \frac{t_0}{k_t}.$$

Model 1:

$$\dot{Nm}_i = \beta_{n0} + \beta_n \cdot \sigma(NOTCH_i \langle DLL_j \rangle, k_p, p) - \alpha_n \cdot Nm_i$$
$$\dot{N}_i = \beta_N \cdot Nm_i - \alpha_N \cdot N_i - N_i \langle D_j \rangle$$
$$\dot{Dm}_i = \beta_{d0} + \beta_d \cdot \delta\left(NOTCH_i \langle DLL \rangle_j, k_d, h\right) - \alpha_d \cdot Dm_i$$
$$\dot{D}_i = \beta_D \cdot Dm_i - \alpha_D \cdot D_i - \langle N_j \rangle D_i$$
$$\dot{R}_i = NOTCH_i \langle DLL \rangle_j - \alpha_R R_i$$

Where:

$$\beta_{n0} \equiv \frac{\beta_{n0}}{t_0 \cdot Nm_0}, \beta_n \equiv \frac{\beta_n}{t_0 \cdot Nm_0}, \beta_N \equiv \frac{\beta_N \cdot Nm_0}{t_0 \cdot N_0}, \beta_{d0} \equiv \frac{\beta_{d0}}{t_0 \cdot Dm_0},$$
$$\beta_d \equiv \frac{\beta_d}{t_0 \cdot Dm_0}, \beta_D \equiv \frac{\beta_D \cdot Dm_0}{t_0 \cdot D_0}, \alpha_n \equiv \frac{\alpha_n}{t_0}, \alpha_N \equiv \frac{\alpha_N}{t_0}, \alpha_d \equiv \frac{\alpha_d}{t_0},$$
$$\alpha_D \equiv \frac{\alpha_D}{t_0}, \alpha_R \equiv \frac{\alpha_R}{t_0}, k_p \equiv \frac{k_p}{k_t \cdot N_0 \cdot D_0}, k_d \equiv \frac{k_d}{k_t \cdot N_0 \cdot D_0}.$$

To modulate the relative strength of the transcriptional regulations (LI, PF), we rescaled the ratios of basal transcriptional rates and regulated transcriptional rates as $S_{PF} \equiv (\beta_n)/(\beta_{n0} + \beta_n)$, $S_{LI} \equiv (\beta_d/(\beta_{d0} + \beta_d))$, and maximum transcriptional rate as $\beta_{nm} = (\beta_{n0} + \beta_n)$, $\beta_{dm} = (\beta_{d0} + \beta_d)$, where $0 \leq S_{PF}, S_{LI} \leq 1$. A new set of equations can be shown as

Model 2:

$$\dot{Nm}_i = \beta_{nm} [(1 - S_{PF}) + S_{PF} \cdot \sigma(NOTCH_i \langle DLL_j \rangle, k_p, p)] - \alpha_n \cdot Nm_i$$
$$\dot{N}_i = \beta_N \cdot Nm_i - \alpha_N \cdot N_i - N_i \langle D_j \rangle$$
$$\dot{Dm}_i = \beta_{dm} [(1 - S_{LI}) + S_{LI} \cdot \delta(NOTCH_i \langle DLL_j \rangle, k_d, h)] - \alpha_d \cdot Dm_i$$
$$\dot{D}_i = \beta_D \cdot Dm_i - \alpha_D \cdot D_i - \langle N_j \rangle D_i$$
$$\dot{R}_i = N_i \langle D_j \rangle - \alpha_R R_i$$

We assume the timescale for mRNA is much faster than proteins, so quasi-steady state method is applied to reduce the mRNA species in the equations:

Set $\dot{Nm}_i = 0, \dot{Dm}_i = 0$

$$Nm_i^* = \frac{\beta_{nm}}{\alpha_n} [(1 - S_{PF}) + S_{PF} \cdot \sigma(NOTCH_i \langle DLL_j \rangle, k_p, p)],$$
$$Dm_i^* = \frac{\beta_{dm}}{\alpha_d} [(1 - S_{LI}) + S_{LI} \cdot \delta(NOTCH_i \langle DLL_j \rangle, k_d, h)]$$

replace $[Nm_i, Dm_i]$ with $[Nm_i^*, Dm_i^*]$ in $\dot{N}_i$ and $\dot{D}_i$ respectively, and set $\beta_N = (\beta_N (\beta_{nm}/\alpha_n))$, $\beta_D = (\beta_D (\beta_{dm}/\alpha_d))$. A simple protein model can be shown as

Model 3:

$$\dot{N}_i = \beta_N \cdot [(1 - S_{PF}) + S_{PF} \cdot \sigma(NOTCH_i \langle DLL_j \rangle, k_p, p)] - \alpha_n \cdot N_i - N_i \langle D_j \rangle$$
$$\dot{D}_i = \beta_D \cdot [(1 - S_{LI}) + S_{LI} \cdot \delta(NOTCH_i \langle DLL_j \rangle, k_d, h)] - \alpha_D \cdot D_i - \langle N_j \rangle D_i$$
$$\dot{R}_i = N_i \langle D_j \rangle - \alpha_R R_i$$

## Computational modeling

The deterministic model was constructed and simulated in MATLAB, and the systems dynamics analysis was solved by numerical optimization in MATLAB. The 3D stochastic crypt model was designed and simulated based on the Glazier–Graner–Hogeweg (GGH) computational model using Compucell3D (Swat *et al*, 2012). A supporting layer covered by a single-cell layer of epithelial cells was designed to mimic the finger-like shape of intestine crypts. All *in silico* epithelial cells inherited an effective energy function programmed in the CGH's cell-lattice configuration to have the desired cell prosperities, behaviors, and interactions. The epithelial cells were programmed to possess the essential cellular prosperities including: (i) cell growth, (ii) cell divisions, (iii) cell–cell adhesion, and (iv) anoikis (when epithelial cells detach from supporting layers). In addition, a module of SBML (Systems Biology Markup Language) solver was applied to integrate the Notch signaling models (LI, PFLI) programmed in SBML (Hucka *et al*, 2003) format to every epithelial cell at the bottom of the crypt. Notch signaling and stochastic cellular dynamics were simulated simultaneously in the combined model. A Notch signaling threshold was assigned to determine Notch$^{high}$ (stem) cells and Notch$^{low}$ (Paneth) cells at the crypt bottom. Notch$^{high}$ cells were programmed to actively grow and divide when their cell volume reached division threshold, while Notch$^{low}$ cells were programmed to neither grew nor divide to mimic differentiated Paneth cells. These cells naturally migrate upward to leave the bottom of the crypt with the force generated by the growing and dividing cells at the crypt base. R was used to analyze and plot the statistical results.

**Expanded View** for this article is available online.

## Acknowledgements

We thank Dr. Spyros Artavanis-Tsakonas for providing the Notch transgenic mice and the Cornell imaging facility. This work was supported by DOD (DARPA): HR0011-16-C-0138, NIH R01GM95990, NIH R01GM114254, NSF 1350659 career award, NSF 1137269, NYSTEM C029543, and CAS Pioneer Hundred Talents Program, the Thousand Young Talents Program of China.

## Author contributions

K-YC, TS, PB, and XS conceived the concept and designed the experiments. TS and K-LT conducted all experiments with murine and human organoids, flow cytometry, and CRISPR editing experiments with assistance from LW, PKLM, SK, NR, MW, JC, and AKV. K-YC and JMB developed the mathematical model together. K-YC performed ChIP-Seq analysis. PB performed the pull-down assay, ChIP-PCR, and additional organoid assays. K-YC, TS, and XS wrote the manuscript with comments from SML, NN, and JAG. All authors approved the paper.

## Conflict of interest

The authors declare that they have no conflict of interest.

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
