## [Review Process File · Molecular Systems Biology]

A Notch positive feedback in the intestinal stem cell niche is essential for stem cell self-renewal

Kai-Yuan Chen, Tara Srinivasan, Kuei-Ling Tung, Julio M. Belmonte, Lihua Wang, Preetish Kadur Lakshminarasimha Murthy, Jiahn Choi, Nikolai Rakhilin, Sarah King, Anastasia Kristine Varanko, Mavee Witherspoon, Nozomi Nishimura, James A. Glazier, Steven Lipkin, Pengcheng Bu, and Xiling Shen

Corresponding author: Xiling Shen, Duke University

Review timeline:	Submission date:	15 September 2016
	Editorial Decision:	19 October 2016
	Revision received:	06 February 2017
	Editorial Decision:	24 February 2017
	Revision received:	26 February 2017
	Editorial Decision:	22 March 2017
	Revision received:	22 March 2017
	Accepted:	27 March 2017

Editor: Maria Polychronidou

Transaction Report:

1st Editorial Decision

19 October 2016

Thank you again for submitting your work to Molecular Systems Biology. We have now heard back from the three referees who agreed to evaluate your manuscript. As you will see below, the reviewers appreciate that the presented findings seem potentially interesting. They raise however a series of concerns, which should be carefully addressed in a revision of the manuscript.

Without repeating all the points listed below, the reviewers think that additional analyses are required to convincingly support the main conclusions and they provide constructive suggestions in this regard.

We have circulated the reports to all reviewers as part of our 'pre-decision cross-commenting' policy. During this process, reviewer #2 questioned the necessity of transplanting organoids in an in vivo model and mentioned that it can be a challenging experiment. As such, and in line with these comments, we think that organoid transplantation experiments are not mandatory for the acceptance of the study. Of course, if available, we would welcome the inclusion of in vivo evidence supporting the Notch positive feedback mechanism.

REFeree REPORTS

Reviewer #1:

Fast dividing intestinal stem cells fuel the process of repetitive tissue renewal. Multiple pathways, including WNT and Notch signals, have been demonstrated to regulate intestinal stem cell self-renewal and differentiation.

Using mouse and human intestinal organoids the authors further characterize the effect of Notch signaling in intestinal Lgr5⁺ cells. They describe a positive feedback mechanism via direct Notch binding to the 2nd intron of the Notch1 gene. Evidence is provided that mutation of this binding site hinders intestinal organoid formation and regeneration.

Comments:

1. The image quality of the lineage tracing data presented in figure EV2 does not match the publication standard. Notch2 derived long term tracing in the intestinal epithelium is described as sporadic units. In fact, the presented image, at 30 days post induction, demonstrates tracing units consisting mostly of only a few RFP positive cells. Given the rapid turnover rate of the intestinal epithelium it seems unlikely that these clones are actually epithelial derived. To verify the origin of such clones a higher image quality is needed. In addition, the tissue may need to be co-stained with epithelial and/or stromal markers.

2. CRISPR-Cas9 induced mutation of the NICD binding motif significantly reduced the colony formation and growth rate of intestinal organoids. According to 2E, however, a number of Lgr5-CBCs (carrying a NICD binding motif mutation) form intestinal organoids. Can such organoids be propagated?

3. Figure EV3G demonstrates that CRISPR-Cas9 induced mutation of the NICD binding motif significantly decreased Notch1 transcript levels in intestinal organoids. In addition, however, it is also shown that JAG1 stimulation rescues the NICD binding motif mutation-induced Notch1 ablation. In fact, such JAG1 treated organoids present an almost 2-fold up-regulated expression level of Notch1 (as compared to non mutated control cells) indicating additional levels of Notch1 regulation. Does JAG1 stimulation also rescue the colony formation and growth rate of intestinal organoids derived from NICD binding motif-mutated cells? If so, do such organoids continue to depend on JAG1 stimulation after propagation? The same questions apply to human colon organoids generated from CRISPR-Cas9 induced mutated cells.

4. The described Notch1 positive feedback loop is overall convincing. However, the entire study is based on intestinal organoids, which may not fully reflect the homeostatic behavior of intestinal stem cells in vivo. To validate the impact of NICD binding motif mutation in vivo, the authors may need to transplant CRISPR-Cas9-derived organoids into superficially damaged mouse colon and characterize the stem cell behavior in engraftment crypts.

Minor comments:

1. The order of the first figure does not correspond to the description in the main text. In addition, 1A is not described in the main text.

2. Figure EV1 demonstrates high levels of Lysozyme expression in sorted CD24^{high} cells. To validate a restricted Lysozyme expression to the population sorted of CD24^{high} cells an immunofluorescence for Lysozyme on sorted Lgr5 cells should be included to this set of data.

3. Human colon Lgr5⁺ cells are described as CBCs. This term, however, only refers to the small intestinal epithelium.

Reviewer #2:

In this manuscript the authors identify a putative enhancer located inside the Notch1 transcribed region that is required to maintain a positive feedback loop for Notch signaling in intestinal stem cells. This is an interesting finding, and in my opinion the experiments are well performed and supportive of the conclusion obtained.

However, writing should be revised for clarity. Moreover, there is too much argumentation in the results section. Authors should try being more precise when they refer exclusively to the results that are shown and include all the interpretation of the data in the discussion section (and the background

in the introduction).

The lineage tracing from N1-cre and N2 Cre is highly overlapping with data published by Fre et al, PlosOne 2011 and to my knowledge does not provide any relevant information that is required to support the message of the work. In contrast other relevant information such as the demonstration that ICN1 directly binds the enhancer region of Notch 1 is only shown in the supplementary figures (EV3C and F). Additional experiments have to be done to further support this essential finding that is in fact the core of the work: I would suggest cloning the enhancer region in reporter vectors and perform quantitative luciferase assays with the wildtype and mutated sequenced (in the RBP sites).

Also, in the text related to the results section, authors should help the readers in understanding what has been done and how. For example "We then designed CRISPR-Cas9 vectors to mutate the NICD/RBPJk binding motif in human colon stem cells (Fig 3C, Fig EV4B). ChIP-qPCR validated that the CRISPR/Cas9- mediated mutation reduced NICD binding to the motif in all three conditions (DMSO, JAG1, and DAPT), and prevented JAG1 treatment from increasing NICD binding (Fig EV4C)". Similar to what happens along the manuscript there is no indication on whether infected cells were selected, what was the frequency of the mutation in the whole population of transduced cells and whether and when the mutation was confirmed by sequencing. All this information is mainly restricted to the figure legends.

The flow-cytometry analysis in Fig 2G is not clear enough as lgr5GFP should produce a well-defined population. Better flow-cytometry analysis and controls for individual staining and isotopic irrelevant antibody are totally required.

Reviewer #3:

In this manuscript, the authors used intestinal organoids to observe a positive feedback, in which NICD promotes its own transcription, in individual intestinal stem cells. The authors argue that this positive feedback helps to generate a robust mosaic pattern of mixed intestinal stem cells and Paneth cells in intestinal niche. They support this conclusion with both experimental and theoretical work. Experimentally, disruption of the Notch receptor transcription disrupts the mosaic pattern. Theoretically, the authors show that incorporating an intracellular positive feedback will increase the speed and robustness of the formation of the mosaic pattern. The arguments in this manuscript are interesting. However, they are not fully supported by the current evidence.

Major questions:

1. The authors use a soluble fragment of the Jag1 protein (sJag1) to activate Notch signaling. This is a highly controversial method and most labs do not report activation using soluble ligands, let alone this short, cysteine rich peptide with unclear fold. The authors need to use an alternative, less controversial method for ligand activation. Furthermore, The data in 1D suggest EDTA and sJag1 have a similar effect on Paneth cells; the Western shows they do not. Based on the Western blot in Fig. 1E, Paneth cells do not express any Notch receptors. How can they be stimulated by addition of the soluble Jag1 peptide?

2. The authors need to stain the cross section for Notch1 expression and for basement membrane proteins. If Notch1 is expressed apically, how is the sJag1 binding to it? Does it enter the organoid? Please provide data to demonstrate the sJag1 can actually access the relevant cells (by adding a fluorophore to the peptide?).

3. The authors used CRISPR to target an intronic enhancer to eliminate the positive feedback. They present multiple sequences in Fig. EV3E, but some of the mutations should not disrupt RBP binding to the site. Which clone was used in the study? Was a pool of CRISPR clones assayed? The authors need to demonstrate loss of binding with EMSA of the predicted site and identify the specific mutation in the PF KO.

Regardless. PF KO led to reduced colony-forming efficiency, reduced growth rate and reduced CBC numbers. On the basis of this data, the authors argue, "the positive feedback promotes CBC self-renewal and maintains the mosaic niche pattern". However, the observed consequence may reflect loss of basal Notch expression. What enhancer regulates basal Notch1 expression? Could that basal enhancer be disrupted as well?

Based on Fig. 1E, Notch2 is insensitive to DAPT whereas Notch1 is. This will suggest that in the organoid Notch1 expression must be induced by Notch2, and therefore, a Notch2 null stem cells will have neither receptor. This is not seen in vivo- could the organoid and the intestinal niche differ in their modes of Notch1 regulation?

And related to this: assuming that the DAPT concentration used in Fig. 1E is used in Fig 2G, how come DAPT cannot block all RBPjk-dsRED activity in vector only cells? If the concentration was not the same, this should be spelled out and justified.

4. The authors use the model simulation assuming a parameter setting supportive of the idea that intracellular positive feedback increases the robustness and speed of mosaic pattern formation. How are the parameters selected? Are they reflecting the natural setting of intestinal niche? What will this model produce with a different set of parameters?

1st Revision - authors' response

06 February 2017

Text continued on next page.

Reviewer #1:

Fast dividing intestinal stem cells fuel the process of repetitive tissue renewal. Multiple pathways, including WNT and Notch signals, have been demonstrated to regulate intestinal stem cell self-renewal and differentiation.

Using mouse and human intestinal organoids the authors further characterize the effect of Notch signaling in intestinal Lgr5+ cells. They describe a positive feedback mechanism via direct Notch binding to the 2nd intron of the Notch1 gene. Evidence is provided that mutation of this binding site hinders intestinal organoid formation and regeneration.

Comments:

1. The image quality of the lineage tracing data presented in figure EV2 does not match the publication standard. Notch2 derived long term tracing in the intestinal epithelium is described as sporadic units. In fact, the presented image, at 30 days post induction, demonstrates tracing units consisting mostly of only a few RFP positive cells. Given the rapid turnover rate of the intestinal epithelium it seems unlikely that these clones are actually epithelial derived. To verify the origin of such clones a higher image quality is needed. In addition, the tissue may need to be co-stained with epithelial and/or stromal markers.

According to Reviewer 2's point 1, similar data have been published by (Fre et al., 2011), who performed Notch2 lineage tracing after 1 day, 3 days, 7 days and 7 months and showed similar sporadic Notch2 labeling from 3 days to 7 months. We have removed this figure and cited the reference.

2. CRISPR-Cas9 induced mutation of the NICD binding motif significantly reduced the colony formation and growth rate of intestinal organoids. According to 2E, however, a number of Lgr5-CBCs (carrying a NICD binding motif mutation) form intestinal organoids. Can such organoids be propagated?

Mutated organoids lacked mature budding structures (Figure 2F), were significantly smaller than the empty vector control organoids, and were extremely difficult to subsequently passage, and could barely propagate.

3. Figure EV3G demonstrates that CRISPR-Cas9 induced mutation of the NICD binding motif significantly decreased Notch1 transcript levels in intestinal organoids. In addition, however, it is also shown that JAG1 stimulation rescues the NICD binding motif mutation-induced Notch1 ablation. In fact, such JAG1 treated organoids present an almost 2-fold up-regulated expression level of Notch1 (as compared to non-mutated control cells) indicating additional levels of Notch1 regulation. Does JAG1 stimulation also rescue the colony formation and growth rate of intestinal organoids derived from NICD binding motif-mutated cells? If so, do such organoids continue to depend on JAG1 stimulation after propagation? The same questions apply to human colon organoids generated from CRISPR-Cas9 induced mutated cells.

We appreciate Reviewer's comment. As a clarification, in Figure EV3G, the RT-qPCR measurement was performed on isolated single cells with Jag1 embedded (premixed) in Matrigel to maximize the interactions between Notch receptors and Jag1 surrounding the single cell, and

such Jag1 treatment lasted 48 hours, as described in the method and figure legend. We have also added this clarification to the revised main text. This might explain the almost 2-fold upregulation. The same setup cannot be applied to testing colony forming efficiency and growth rate, which requires continuous longer-term organoid culturing without breaking Matrigel to embed JAG1. Simply adding soluble JAG1 to the medium did not rescue colony formation or growth rates.

To investigate whether additional Notch activation can rescue colony formation, we performed the rescue experiment using a Tamoxifen-inducible LGR5-EGFP-CreERT2 x Rosa26-YFP-NICD (NICD-OE) mouse strain (Oh et al., 2013). Single LGR5/NICD-OE intestine cells were transfected with either empty vector or CRISPR/Cas9 gRNAs. Organoids under normal condition (w/o tamoxifen induction) and under tamoxifen treatment were compared (Figure EV3). The colony forming efficiency in CRISPR/Cas9 mutated organoids was rescued by Tamoxifen induction that activates NICD expression, while organoid sizes were only partially rescued. This suggests that ectopic Notch signaling rescues self-renewal, but Notch positive feedback may play an additional role in organoid growth.

4. The described Notch1 positive feedback loop is overall convincing. However, the entire study is based on intestinal organoids, which may not fully reflect the homeostatic behavior of intestinal stem cells *in vivo*. To validate the impact of NICD binding motif mutation *in vivo*, the authors may need to transplant CRISPR-Cas9-derived organoids into superficially damaged mouse colon and characterize the stem cell behavior in engraftment crypts.

We acknowledge the potential difference between organoids and crypts *in vivo*, which receive additional signaling cues from the underlying mesenchyme. However, engraftment of mouse colon organoids via enema has been published once (Yui et al., 2012) with few successful follow-ups, and the experts we have consulted think that the engraftment efficiency is extremely low. Given the low self-renewal and viability of the mutated organoids, the chance of successful engraftment is probably too low. Furthermore, all the data in this study was based on mouse intestinal organoids and human colon organoids, but not on mouse colon organoids. The editor has advised us not to pursue this particular experiment.

Minor comments:

1. The order of the first figure does not correspond to the description in the main text. In addition, 1A is not described in the main text.

We thank Reviewer for the careful reading. We have revised the manuscript accordingly.

2. Figure EV1 demonstrates high levels of Lysozyme expression in sorted CD24^{high} cells. To validate a restricted Lysozyme expression to the population sorted of CD24^{high} cells an immunofluorescence for Lysozyme on sorted Lgr5 cells should be included to this set of data.

We performed the suggested experiment. Lysozyme was expressed in sorted CD24^{high} cells, but not in LGR5-GFP cells (Figure EV1A).

3. Human colon Lgr5⁺ cells are described as CBCs. This term, however, only refers to the

small intestinal epithelium.

We have corrected the description accordingly.

Reviewer #2:

In this manuscript the authors identify a putative enhancer located inside the Notch1 transcribed region that is required to maintain a positive feedback loop for Notch signaling in intestinal stem cells. This is an interesting finding, and in my opinion the experiments are well performed and supportive of the conclusion obtained.

However, writing should be revised for clarity. Moreover, there is too much argumentation in the results section. Authors should try being more precise when they refer exclusively to the results that are shown and include all the interpretation of the data in the discussion section (and the background in the introduction).

We appreciate Reviewer's advice on rigorous scientific writing and have revised the manuscript accordingly.

1. The lineage tracing from N1-cre and N2 Cre is highly overlapping with data published by Fre et al, PlosOne 2011 and to my knowledge does not provide any relevant information that is required to support the message of the work.

We have removed the data following Reviewer's advice. As Reviewer pointed out, our data were consistent with Fre et al. The only difference was that we used fluorescent reporters instead of X-gal staining and β -galactosidase.

2. In contrast other relevant information such as the demonstration that ICN1 directly binds the enhancer region of Notch 1 is only shown in the supplementary figures (EV3C and F). Additional experiments have to be done to further support this essential finding that is in fact the core of the work: I would suggest cloning the enhancer region in reporter vectors and perform quantitative luciferase assays with the wildtype and mutated sequenced (in the RBP sites).

We appreciate Reviewer's excellent suggestion and performed the luciferase reporter assay as suggests (Figure 2C). The wild-type and three mutated binding sequences were cloned into the enhancer sites of the pGL4.27 [luc2P/minP] luciferase reporter vector with a minimal promoter (Figure 2C, left). The luciferase signals from the 3 mutated sequences and the control sequence (without the binding site) were significantly lower than the wild-type sequence (Figure 2C). The wild-type sequence responded most strongly to Jag1 and DAPT treatments, consistent with the core message.

Furthermore, we also performed a pull-down assay to validate the binding. Briefly, we synthesized oligonucleotides of the wild-type and mutated enhancer sequences labeled with biotin to pull down NICD/RBP complex from mouse intestinal crypt cell lysates. The interaction of the enhancer and NICD/RBP complex were validated by Western blot (Figure 2D). The pull-down assay confirms that NICD interacts with the identified binding sequence, which is

consistent with the luciferase assay.

3. Also, in the text related to the results section, authors should help the readers in understanding what has been done and how. For example, "We then designed CRISPR-Cas9 vectors to mutate the NICD/RBPJk binding motif in human colon stem cells (Fig 3C, Fig EV4B). ChIP-qPCR validated that the CRISPR/Cas9- mediated mutation reduced NICD binding to the motif in all three conditions (DMSO, JAG1, and DAPT), and prevented JAG1 treatment from increasing NICD binding (Fig EV4C) ". Similar to what happens along the manuscript there is no indication on whether infected cells were selected, what was the frequency of the mutation in the whole population of transduced cells and whether and when the mutation was confirmed by sequencing. All this information is mainly restricted to the figure legends.

We have included more description in the revised manuscript. Briefly, after transfection of the CRISPR/Cas9 plasmid and puromycin selection, selected clones were picked and sequenced respectively. CRISPR/Cas9 was shown to have high efficiency of gene editing through NHEJ (Non-Homologous End Joining) and inducing indels on 3-4 bases upstream to the PAM (Protospacer Admacent Motif) sequence (Cong et al., 2013; Hsu et al., 2013; Wang et al., 2013; Yang et al., 2013). Our designed sgRNAs overlap with the motif sequence created indels of 0-3 bases, which are 3-4 bases upstream to PAM sequence, consistent with other CRISPR editing papers.

4. The flow-cytometry analysis in Fig 2G is not clear enough as lgr5GFP should produce a well-defined population. Better flow-cytometry analysis and controls for individual staining and isotopic irrelevant antibody are totally required.

The mentioned flow-cytometry analysis shows the distribution of both LGR5-EGFP and RBPJK-dsRED levels. We re-plotted the same data based on LGR5-EGFP alone, which showed a more defined population (Rebuttal Figure1, top).

Rebuttal Figure 1. Single empty vector control or CRISPR/Cas9-positive feedback knockout (PF KO) LGR5-EGFP+ CBCs were transfected with an RBPJk-dsRed reporter construct and grown into organoids, which were subsequently treated with DMSO, DAPT or JAG1 for 48 hours. UP: Representative FACS plots for FSC and LGR5-EGFP expression indicating a gated positive fraction for each condition. Bottom: Representative FACS plots for RBPJk-dsRed expression of the entire cell population.

Reviewer #3:

In this manuscript, the authors used intestinal organoids to observe a positive feedback, in which N1ICD promotes its own transcription, in individual intestinal stem cells. The authors argue that this positive feedback helps to generate a robust mosaic pattern of mixed intestinal stem cells and Paneth cells in intestinal niche. They support this conclusion with both experimental and theoretical work. Experimentally, disruption of the Notch receptor transcription disrupts the mosaic pattern. Theoretically, the authors show that incorporating an intracellular positive feedback will increase the speed and robustness of the formation of the mosaic pattern.

The arguments in this manuscript are interesting. However, they are not fully supported by the current evidence.

Major questions:

1. The authors use a soluble fragment of the Jag1 protein (sJag1) to activate Notch signaling. This is a highly controversial method and most labs do not report activation using soluble ligands, let alone this short, cysteine rich peptide with unclear fold. The authors need to use an alternative, less controversial method for ligand activation. Furthermore, the data in 1D suggest EDTA and sJag1 have a similar effect on Paneth cells; the Western shows they do not. Based on the Western blot in Fig. 1E, Paneth cells do not express any Notch receptors. How can they be stimulated by addition of the soluble Jag1 peptide?

There have been quite a few reports showing that recombinant Jag1 activates Notch signaling

(Sainson et al., 2005; Xie et al., 2015; Yamamura et al., 2014). Recombinant Jag1 has also been used in intestinal organoid in several papers (Mahapatro et al., 2016; Sato et al., 2009; Srinivasan et al., 2016; Van Landeghem et al., 2012; VanDussen et al., 2012). Our Jag1 results seem to be consistent with activation by EDTA as well (Rand et al., 2000).

However, we were fully aware of the controversy given that Notch ligands are thought to act via mechanical pulling—hence ligand hybridization to a surface seems to be required (Gordon et al., 2015; Wang and Ha, 2013). We found that Jag1 only worked when being pre-mixed with Matrigel, potentially because the embedment in the Matrigel provides Jag1 with some stabilizing force to activate Notch receptors. Notch stimulation by Jag1 embedded in Matrigel was confirmed in multiple assays including RT-qPCR (Figure 1D), western blot (Figure 1E), and RBPjk reporter (Figure 2I). On the other hand, adding Jag1 directly to the medium does not activate Notch signaling, consistent with Reviewer’s concern (Rebuttal Figure 2). Besides the figure legends, we added this clarification to the main text.

Rebuttal Figure 2. RT-qPCR quantification of Notch signaling components in intestine stem cells after organoids were either treated with free JAG1 in culturing medium or without adding Jag1 (ctrl). The experiments were performed in triplicate and presented mean \pm S.T.D.

Paneth cells may still have low baseline Notch receptor and effector transcript levels (e.g., most promoters are somewhat leaky even when they are off) detectable by sensitive RT-qPCR in Figure 1D, which only shows relatively fold change. However, the receptor and effector protein levels are below detection threshold for Western blot in Figure 1E.

2. The authors need to stain the cross section for Notch1 expression and for basement membrane proteins. If Notch1 is expressed apically, how is the sJag1 binding to it? Does it enter the organoid? Please provide data to demonstrate the sJag1 can actually access the relevant cells (by adding a fluorophore to the peptide?).

We followed Reviewer’s suggestion and stained the intestine crypt for Notch1, which showed alternating Notch1 pattern at the crypt bottom (Rebuttal Figure 3), consistent with images of Notch1 reporter (Figure 2A/B) and western blotting (Figure 1E). Notch1 is expressed on apical, lateral, and basal sides. Furthermore, as explain previously, in our JAG1 experiment, dissociated single cells were seeded in Matrigel pre-mixed with Jag1 for 48 hours, so Notch1 receptors on all sides of the single cells are accessible to embedded sJag1 in Matrigel.

Rebuttal Figure 3. Immunofluorescent images of intestine crypts. Red: Notch1, Blue: DAPI. Scale bar represents 50 μm .

3. The authors used CRISPR to target an intronic enhancer to eliminate the positive feedback. They present multiple sequences in Fig. EV3E, but some of the mutations should not disrupt RBP binding to the site. Which clone was used in the study? Was a pool of CRISPR clones assayed? The authors need to demonstrate loss of binding with EMSA of the predicted site and identify the specific mutation in the PF KO.

As a clarification, after transfection of the CRISPR/Cas9 plasmid and puromycin selection, selected clones were picked and sequenced separately. For the study, a pool of CRISPR clones were assayed. We have added this information to the revised manuscript.

To answer the questions of binding efficiency, we first performed a luciferase reporter assay as suggested by Reviewer 2. The wildtype and three mutated binding sequences were cloned to the enhancer sites of the pGL4.27 [luc2P/minP] luciferase reporter vector with a minimal promoter (Figure 2C). The luciferase signals from the 3 mutated sequences and the control sequence (without the binding site) were significantly lower than the wild-type sequence (Figure 2C). The wild-type sequence responded most strongly to Jag1 and DAPT treatments, consistent with the core message.

As reviewer suggests, we also performed additional experiments to confirm binding. In the EMSA assay, only recombinant NICD and RBP proteins would be added, which may lack other necessary co-factors present in intestinal cells. Instead, we performed a pull-down assay to verify NICD binding on the sequence. Briefly, we synthesized oligonucleotides of the wild-type and mutated enhancer sequences (including total replacement of motif sequence, mutations on the CRISPR cutting sites and flanking region, and indels around the cutting site, see Figure 2D), labeled the enhancer DNA with biotin, and use the biotin label to pull down enhancer-binding NICD/RBPjk complex from mouse intestinal lysates. Western blot validated the interaction between the enhancer and NICD/RBP complex (Figure 2D). The WT sequence showed strongest NICD binding, while the mutated sequences had weaker binding. The pulldown assay is consistent with the luciferase reporter assay (Figure 2C) and the ChIP assay (Figure EV2F).

We thank Reviewer for the excellent suggestions that helped strengthen the core finding of this paper.

4. Regardless. PF KO led to reduced colony-forming efficiency, reduced growth rate and reduced CBC numbers. On the basis of this data, the authors argue, "the positive feedback promotes CBC self-renewal and maintains the mosaic niche pattern". However, the observed

consequence may reflect loss of basal Notch expression. What enhancer regulates basal Notch1 expression? Could that basal enhancer be disrupted as well?

Currently it is not known which enhancer region(s) regulate basal Notch1 expression in intestine stem cells. The NICD-binding enhancer we identified is in the second intron, which is not a most likely location for regulating basal Notch1 expression. Furthermore, the luciferase reporter experiment confirmed that the activity of this enhancer site responds to Notch signaling level rather than acting as a basal site (Figure 2C).

Our lab has started a new project to epigenetically profile intestinal stem cells using ATAC-Seq and ChIP-Seq followed by a CRISPR screening to identify specific epigenetic regions that affect the expression of Notch genes. We hope to provide more answers to this intriguing question in the future.

5. Based on Fig. 1E, Notch2 is insensitive to DAPT whereas Notch1 is. This will suggest that in the organoid Notch1 expression must be induced by Notch2, and therefore, a Notch2 null stem cells will have neither receptor. This is not seen *in vivo*- could the organoid and the intestinal niche differ in their modes of Notch1 regulation?

We agree with the Reviewer that Notch2 is not required to induce Notch1 *in vivo*, as Notch1 inhibition causes a defective intestinal phenotype, while Notch2 inhibition causes no significant phenotype (Wu et al., 2010). The organoid does not have mesenchyme, which may provide additional signaling cues. One example is that Paneth cells are essential for organoids but dispensable *in vivo*, as the mesenchyme can provide Wnt ligands in place of Paneth cells (Durand et al., 2012).

6. And related to this: assuming that the DAPT concentration used in Fig. 1E is used in Fig 2G, how come DAPT cannot block all RBPjk-dsRED activity in vector only cells? If the concentration was not the same, this should be spelled out and justified.

The Western blot and FACS of the RBPjk-dsRed reporter cells showed similar trends. However, it may be difficult to compare them quantitatively. First, the reporter may be more sensitive than blotting, as it has 12 tandem binding sites. Second, Western blot measures the average and depends on the exposure time.

7. The authors use the model simulation assuming a parameter setting supportive of the idea that intracellular positive feedback increases the robustness and speed of mosaic pattern formation. How are the parameters selected? Are they reflecting the natural setting of intestinal niche? What will this model produce with a different set of parameters?

The parameters of the model are based on previous Notch models from the Elowitz lab (Sprinzak et al., 2011; Sprinzak et al., 2010). The parameters are also varied to explore different system properties in the parameter spaces (as shown in Figure 4B,C, Figure EV 5A-D), which showed either patterning or non-patterning behaviors at the population level and either bistability or monostability at the single cell level.

We acknowledge that the computational model is unlikely to capture all possible scenarios and certainly cannot “prove” that positive feedback increases robustness. After all, all models are wrong but some models are useful. The computational analysis merely provides a likely explanation as to why the positive feedback might be useful to intestinal stem cells.

Reference:

- Cong, L., Ran, F.A., Cox, D., Lin, S., Barretto, R., Habib, N., Hsu, P.D., Wu, X., Jiang, W., Marraffini, L.A., *et al.* (2013). Multiplex genome engineering using CRISPR/Cas systems. *Science* 339, 819-823.
- Durand, A., Donahue, B., Peignon, G., Letourneur, F., Cagnard, N., Slomianny, C., Perret, C., Shroyer, N.F., and Romagnolo, B. (2012). Functional intestinal stem cells after Paneth cell ablation induced by the loss of transcription factor Math1 (Atoh1). *Proc Natl Acad Sci U S A* 109, 8965-8970.
- Fre, S., Hannezo, E., Sale, S., Huyghe, M., Lafkas, D., Kissel, H., Louvi, A., Greve, J., Louvard, D., and Artavanis-Tsakonas, S. (2011). Notch lineages and activity in intestinal stem cells determined by a new set of knock-in mice. *PLoS one* 6, e25785-e25785.
- Gordon, W.R., Zimmerman, B., He, L., Miles, L.J., Huang, J., Tiyanont, K., McArthur, D.G., Aster, J.C., Perrimon, N., Loparo, J.J., *et al.* (2015). Mechanical Allostery: Evidence for a Force Requirement in the Proteolytic Activation of Notch. *Dev Cell* 33, 729-736.
- Hsu, P.D., Scott, D.A., Weinstein, J.A., Ran, F.A., Konermann, S., Agarwala, V., Li, Y., Fine, E.J., Wu, X., Shalem, O., *et al.* (2013). DNA targeting specificity of RNA-guided Cas9 nucleases. *Nat Biotechnol* 31, 827-832.
- Mahapatro, M., Foersch, S., Hefele, M., He, G.W., Giner-Ventura, E., McHedlidze, T., Kindermann, M., Vetrano, S., Danese, S., Gunther, C., *et al.* (2016). Programming of Intestinal Epithelial Differentiation by IL-33 Derived from Pericryptal Fibroblasts in Response to Systemic Infection. *Cell reports* 15, 1743-1756.
- Oh, P., Lobry, C., Gao, J., Tikhonova, A., Loizou, E., Manent, J., van Handel, B., Ibrahim, S., Greve, J., Mikkola, H., *et al.* (2013). In vivo mapping of notch pathway activity in normal and stress hematopoiesis. *Cell stem cell* 13, 190-204.
- Rand, M.D., Grimm, L.M., Artavanis-Tsakonas, S., Patriub, V., Blacklow, S.C., Sklar, J., and Aster, J.C. (2000). Calcium depletion dissociates and activates heterodimeric notch receptors. *Mol Cell Biol* 20, 1825-1835.
- Sainson, R.C., Aoto, J., Nakatsu, M.N., Holderfield, M., Conn, E., Koller, E., and Hughes, C.C. (2005). Cell-autonomous notch signaling regulates endothelial cell branching and proliferation during vascular tubulogenesis. *FASEB J* 19, 1027-1029.
- Sato, T., Vries, R.G., Snippert, H.J., van de Wetering, M., Barker, N., Stange, D.E., van Es, J.H., Abo, A., Kujala, P., Peters, P.J., *et al.* (2009). Single Lgr5 stem cells build crypt-villus structures in vitro without a mesenchymal niche. *Nature* 459, 262-265.
- Sprinzak, D., Lakhanpal, A., LeBon, L., Garcia-Ojalvo, J., and Elowitz, M.B. (2011). Mutual inactivation of Notch receptors and ligands facilitates developmental patterning. *PLoS Comput Biol* 7, e1002069.
- Sprinzak, D., Lakhanpal, A., Lebon, L., Santat, L.A., Fontes, M.E., Anderson, G.A., Garcia-Ojalvo, J., and Elowitz, M.B. (2010). Cis-interactions between Notch and Delta generate mutually exclusive signalling states. *Nature* 465, 86-90.

Srinivasan, T., Than, E.B., Bu, P., Tung, K.L., Chen, K.Y., Augenlicht, L., Lipkin, S.M., and Shen, X. (2016). Notch signalling regulates asymmetric division and inter-conversion between *Lgr5* and *bmi1* expressing intestinal stem cells. *Sci Rep* 6, 26069.

Van Landeghem, L., Santoro, M.A., Krebs, A.E., Mah, A.T., Dehmer, J.J., Gracz, A.D., Scull, B.P., McNaughton, K., Magness, S.T., and Lund, P.K. (2012). Activation of two distinct *Sox9*-EGFP-expressing intestinal stem cell populations during crypt regeneration after irradiation. *American journal of physiology Gastrointestinal and liver physiology* 302, G1111-1132.

VanDussen, K.L., Carulli, A.J., Keeley, T.M., Patel, S.R., Puthoff, B.J., Magness, S.T., Tran, I.T., Maillard, I., Siebel, C., Kolterud, A., *et al.* (2012). Notch signaling modulates proliferation and differentiation of intestinal crypt base columnar stem cells. *Development (Cambridge, England)* 139, 488-497.

Wang, H., Yang, H., Shivalila, C.S., Dawlaty, M.M., Cheng, A.W., Zhang, F., and Jaenisch, R. (2013). One-step generation of mice carrying mutations in multiple genes by CRISPR/Cas-mediated genome engineering. *Cell* 153, 910-918.

Wang, X., and Ha, T. (2013). Defining single molecular forces required to activate integrin and notch signaling. *Science* 340, 991-994.

Wu, Y., Cain-Hom, C., Choy, L., Hagenbeek, T.J., de Leon, G.P., Chen, Y., Finkle, D., Venook, R., Wu, X., Ridgway, J., *et al.* (2010). Therapeutic antibody targeting of individual Notch receptors. *Nature* 464, 1052-1057.

Xie, K., Qiao, F., Sun, Y., Wang, G., and Hou, L. (2015). Notch signaling activation is critical to the development of neuropathic pain. *BMC Anesthesiol* 15, 41.

Yamamura, H., Yamamura, A., Ko, E.A., Pohl, N.M., Smith, K.A., Zeifman, A., Powell, F.L., Thistlethwaite, P.A., and Yuan, J.X. (2014). Activation of Notch signaling by short-term treatment with Jagged-1 enhances store-operated Ca(2+) entry in human pulmonary arterial smooth muscle cells. *American journal of physiology Cell physiology* 306, C871-878.

Yang, H., Wang, H., Shivalila, C.S., Cheng, A.W., Shi, L., and Jaenisch, R. (2013). One-step generation of mice carrying reporter and conditional alleles by CRISPR/Cas-mediated genome engineering. *Cell* 154, 1370-1379.

Yui, S., Nakamura, T., Sato, T., Nemoto, Y., Mizutani, T., Zheng, X., Ichinose, S., Nagaishi, T., Okamoto, R., Tsuchiya, K., *et al.* (2012). Functional engraftment of colon epithelium expanded in vitro from a single adult *Lgr5*(+) stem cell. *Nat Med* 18, 618-623.

Thank you again for submitting your work to Molecular Systems Biology. We have now heard back from the two referees who agreed to evaluate your manuscript. As you will see below, reviewer #3 still raises some remaining issues that we would ask you to address in a revision of the manuscript.

Reviewer #3 thinks (point #1) that additional experiments, in which an alternative approach (additionally to Jag1) is used for activating Notch signaling, should be included. During our pre-precision cross-commenting process, in which all referees see and can comment on each others' reports, reviewer #2 mentioned: "Although I agree with the author's response to comment #1 of reviewer #3 (and in my lab we have also been able to induce Notch activation by soluble Jag1 in matrigel-growing tumoroids), I would suggest accepting the manuscript by demanding what reviewer 3 proposes as possibility 2 (discussing the issue in the manuscript). As such, we think that it is not mandatory to include additional experiments, but it is sufficient to discuss this issue in the text.

REFEREE REPORTS

Reviewer #2:

In my opinion, the manuscript in its revised form is now acceptable for publication.

Reviewer #3:

In general, the authors chose to debate our comments instead of addressing them experimentally. They acknowledged that sJag is a controversial reagent acting via an unknown mechanism, requiring matrigel, and thus signaling through ill-defined pathways that may or may not include direct activation of Notch. Likewise, the authors made an effort to conjure up an explanation for why two different methods of Notch stimulation, sJag1 and EDTA, produced different results in Figure 1E, without acknowledgment that the difference could reflect sJag1 ability to activate some other pathway(s) which then activate Notch. We asked that they buttress their conclusions using a second method. By not performing the experiment, the authors left their main point with only partial support.

We suggests two possible routes:

1. The authors will carry out the suggested experiments to remove ambiguity so that the manuscript is either strongly supported or modified according to the new results.
2. Alternatively, the editors should require that they openly discuss the caveats we raised within the manuscript, so that the readers can make their own judgments.

It is up to the authors and editors to decide which route should be taken.

In reaction to their responses, we wish to make the following additional comments:

- Point1: The image resolution makes it hard to judge the precise positions of Notch in membranes.
- Point 2: it is hard for the reviewer to see how the findings with dissociated cells support the author's main argument on pattern formation within the multicellular organ.
- Point 5 - this needs to be discussed in the manuscript.
- Point 6: The authors attribute the different observations to the difference between single cells and population of cells. This hypothesis should be rigorously tested with the aid of mathematical modeling.
- Point 7: These should be openly discussed in the manuscript, and the tone of the manuscript adjusted accordingly.

2nd Revision - authors' response

26 February 2017

We have addressed Reviewer 3's concern by adding a paragraph describing the caveats of using JAG1 at the end of the Discussion section.

3rd Editorial Decision

22 March 2017

Thank you again for sending us your revised manuscript. We think that the remaining issues have been satisfactorily addressed and the manuscript is now suitable for publication.

Corresponding Author Name: Dr. Xiling Shen, Dr. Pengcheng Bu

Manuscript Number: MSB-16-7324